# The northern European shelf as increasing net sink for $CO_2$

Meike Becker[1,2], Are Olsen[1,2], Peter Landschützer[3], Abdirhaman Omar[4,2], Gregor Rehder[5], Christian Rödenbeck[6], and Ingunn Skjelvan[4,2]

[1]Geophysical Institute, University of Bergen, Bergen, Norway
[2]Bjerknes Center for Climate Research, Bergen, Norway
[3]Max Planck Institute for Meteorology, Hamburg, Germany
[4]NORCE Norwegian Research Centre AS, Bergen, Norway
[5]Leibniz Institute for Baltic Sea Reasearch, Warnemünde, Germany
[6]Max Planck Institute for Biogeochemistry, Jena, Germany

**Correspondence:** Meike Becker (meike.becker@uib.no)

**Abstract.** We developed a simple method to refine existing open ocean maps and extending them towards different coastal seas. Using a multi linear regression we produced monthly maps of surface ocean $fCO_2$ in the northern European coastal seas (North Sea, Baltic Sea, Norwegian Coast and in the Barents Sea) covering a time period from 1998 to 2016. A comparison with gridded SOCAT v5 data revealed mean biases and standard deviations of $0\pm26\mu$atm in the North Sea, $0\pm16\mu$atm along the Norwegian Coast, $0\pm19\mu$atm in the Barents Sea, and $2\pm42\mu$atm in the Baltic Sea. We used these maps to investigate trends in $fCO_2$, pH and air-sea $CO_2$ flux. The surface ocean $fCO_2$ trends are smaller than the atmospheric trend in most of the studied regions. The only exception to this is the western part of the North Sea, where sea surface $fCO_2$ increase by 2 $\mu$atm yr$^{-1}$, which is similar to the atmospheric trend. The Baltic Sea does not show a significant trend. Here, the variability was much larger than the expected trends. Consistently, the pH trends were smaller than expected for an increase of $fCO_2$ in pace with the rise of atmospheric $CO_2$ levels. The calculated air-sea $CO_2$ fluxes revealed that most regions were net sinks for $CO_2$. Only the southern North Sea and the Baltic Sea emitted $CO_2$ to the atmosphere. Especially in the northern regions the sink strength increased during the studied period.

## 1 Introduction

For facing global challenges, such as predicting and tracking climate change, it is important to improve our understanding of the ocean carbon sink and its variability. Open oceans, especially those of the northern hemisphere, are relatively well understood and described in their large-scale variability (Gruber et al., 2019; Landschützer et al., 2018, 2019; Fay and McKinley, 2017). Reliable autonomous systems for measuring carbon dioxide partial pressure from commercial vessels were developed in the early 2000s (Pierrot et al., 2009) and have since been deployed on a large number of vessels (e.g. Bakker et al. (2016)). This has resulted in sufficient data to develop methods to interpolate the data and to describe large scale air-sea $CO_2$ exchange and

its variability (Landschützer et al., 2014, 2013; Rödenbeck et al., 2013; Jones et al., 2015). These methods apply wide variety of approaches, such as linear interpolation, machine learning, and model based estimates. By comparing the different results, it is possible to achieve a good estimate of the uncertainty associated with the respective methods (Rödenbeck et al., 2015).

Despite coastal seas cover 7-10% of the world's oceans (Bourgeois et al., 2016), their contribution to the oceanic carbon sink is not yet fully constrained. Whether coastal seas are a net sink or source for atmospheric $CO_2$ and how their role will change in a changing climate is still under debate (Bauer et al., 2013; Laruelle et al., 2010). Compared to the open ocean, longer time series and higher spatial and temporal resolution of the observations are needed in order to capture all relevant coastal processes. Small scale circulation patterns governed by topographic features, thermal and haline stratification, or mixing through tidal cycles, upwelling or internal waves result in a need for more complex maps with a higher resolution (Bricheno et al., 2014; Lima and Wethey, 2012; Blanton, 1991). These physical drivers are not the only reasons for coastal seas being more complicated to understand. Generally, coastal regions are more productive than open ocean regions due to better availability of nutrients (e.g. mixing at continental margins, river runoff). While deeper coastal regions are seasonally stratified, shallow regions are vertically mixed allowing for exchange between the benthic and pelagic parts of the ecosystem (Griffiths et al., 2017; Wollast, 1998). Together with strong gradients of productivity this leads to spatial and temporal heterogeneity in surface $CO_2$ content.

The different maps developed for describing the open ocean surface $pCO_2$ ($CO_2$ partial pressure) dynamics and air-sea $CO_2$ fluxes are not directly applicable in coastal regions. Many exclude data from continental shelves completely while all of them have too coarse spatial resolution (typically between 1 and 5 °) to properly resolve coastal seas with their small-scale variability. A few studies have described coastal carbon dynamics but most of them have strong regional or temporal limitations. Table 1 shows an overview of studies with estimated $pCO_2$ trends over the northern European shelf while Table 2 presents available flux estimates. Laruelle et al. (2017) used a neural network approach to produce a global $pCO_2$ climatology for coastal seas, describing more distinct seasonal variability in the northern hemisphere than in the southern Pacific and Atlantic. A global climatology covering both open ocean and coastal regions was recently constructed by combining this product with the open ocean product of Landschützer et al. (2016) (Landschützer et al., 2020). Laruelle et al. (2018) published trend estimates in regions with high data coverage based on winter data spanning up to 35 years. Their findings is that the $pCO_2$ rise in coastal regions tend to lag the atmospheric rise in $CO_2$. However, few studies attempted to constrain coastal air-sea fluxes before. Kitidis et al. (2019) estimated fluxes between 0 and -15 mmol m$^{-2}$ day$^{-1}$ in the North Sea, depending on the season (more negative during summer than during winter) and the region (more negative fluxes in the northern North Sea compared to the south). For the Baltic Sea, Parard et al. (2016, 2017) used a neural network approach to produce surface ocean $pCO_2$ maps from 1998 to 2011 and estimated an air-sea flux of 1.2 mmol m$^{-2}$ day$^{-1}$. Yasunaka et al. (2018) estimated a flux of 8 - 12 mmol m$^{-2}$ day$^{-1}$ in the Barents Sea and along the Norwegian coast using a self-organizing map technique. Most of the other available studies on the trends in coastal $pCO_2$ are based on data from either summer or winter. Estimates based on summer-only data typically show large interannual variations (Thomas et al., 2007; Salt et al., 2013), which led to the conclusion that here the interannual variability masks the actual long term trend. The approach to use winter-only data (Fröb et al., 2019; Omar et al., 2019), however, is based on the assumption that during this season the influence of biological processes is negligible

**Table 1.** Overview of trends in surface ocean $CO_2$ reported in the literature.

| | Reference | Time range | $dp CO_2 / dt$ /$\mu$atm yr$^{-1}$ |
|---|---|---|---|
| North Sea | Thomas et al. (2007) | 2001-2005, summer data normalized to 16° | 7.9 |
| North Sea | Salt et al. (2013) | 2001-2005, summer data, normalized to 16° | 6.5 |
| North Sea | Salt et al. (2013) | 2005-2008, summer data, normalized to 16° | 1.33 |
| Faeroe Banks | Fröb et al. (2019) | 2004-2017, winter data (DJFM) | $2.25 \pm 0.20$ |
| North Sea, west | Omar et al. (2019) | 2004-2017, winter data (DJ) | $2.19 \pm 0.55$ |
| North Sea, east | Omar et al. (2019) | 2004-2017, winter data (DJF) | not significant |
| North Sea | Laruelle et al. (2018) | 1988-2015 | almost no trend |
| English channel | Laruelle et al. (2018) | 1988-2015 | slightly smaller than atmosphere |
| Baltic Sea | Wesslander et al. (2010) | 1994-2008 | larger than atmosphere |
| Baltic Sea | Schneider and Müller (2018) | 2008-2015 | 4.6 - 6.1 |
| Baltic Sea, west | Laruelle et al. (2018) | 1988-2015 | much smaller than atmosphere, slightly negative |
| Barents Sea | Yasunaka et al. (2018) | 1997-2013 | not significant |
| Barents Sea | Laruelle et al. (2018) | 1988-2015 | about the same as atmosphere |
| Atmosphere | global average | 1997-2016 | 2.02 ppm yr$^{-1}$ |

**Table 2.** Overview of air-sea $CO_2$ fluxes reported in the literature. Negative sign denotes flux from atmosphere to ocean.

| | Reference | Time range | F / mmol m$^{-2}$ day$^{-1}$ |
|---|---|---|---|
| North Sea | Meyer et al. (2018) | 2001/2002 | -3.8 |
| North Sea | Kitidis et al. (2019) | 2015 | 0 - -15 |
| Baltic Sea | Parard et al. (2017) | 1998-2011 | 1.2 |
| Norwegian Coast | Yasunaka et al. (2018) | 1997-2013 | -4 - -8 |
| Barents Sea | Yasunaka et al. (2018) | 1997-2013 | -8 - -12 |

and therefore winter data can be used to establish a baseline trend. However, also using winter-only data has its drawbacks. In particular the choice of which months to include can cause biases and the optimal selection can differ from region to region.

In this study we present a new approach to develop monthly $f CO_2$ ($CO_2$ fugacity) maps based on already existing open ocean $p CO_2$ maps, in four example regions: North Sea, Baltic Sea, Norwegian Coast and the Barents Sea. A multi linear regression (MLR) was used to fit driver data against $f CO_2$ observations. Based on the resulting $f CO_2$ maps and a salinity-alkalinity correlation we also produced monthly maps of coastal pH. The performance of the produced maps was evaluated

and the maps were then used to investigate trends in coastal $f\mathrm{CO}_2$ and pH in the entire region from 1998 to 2016. Finally, we used the $f\mathrm{CO}_2$ maps to determine the air-sea $\mathrm{CO}_2$ exchange and its temporal and spatial patterns.

## 2 Method

### 2.1 Study area

This work focuses on the northern European continental shelf and marginal seas. As we want to show the performance of the MLR method we picked a number of regions with very different characteristics: the North Sea, the Baltic Sea, the Norwegian coast and the western Barents Sea (Figure 1). We decided to concentrate on these regions because (1) the data coverage in these regions is fairly high and (2) the authors have strong knowledge on the specific regions. This is important in order to properly evaluate the maps and to assess whether or not the output is realistic. The four regions were defined based on the COastal

Segmentation and related CATchments (COSCAT) segmentation scheme (Laruelle et al., 2013). The threshold for defining a region as coastal sea was set to a depth limit of 500 m. By using this definition, we produce an overlap to the open ocean maps, allowing our maps to be merged with the open ocean maps. Please note, that this study concentrates on the continental shelf area. The near coastal zones (e.g. intertidal zones) are not included due to the limited availability of driver data in these regions.

### 2.2 Data handling

The $\mathrm{CO}_2$ data used in this study were extracted from SOCAT version 5 (Bakker et al., 2016). Their coverage is shown in Figure 2. A newer version of the SOCAT database (SOCATv2019) was used for validating the maps against independent data. An overview over the reanalysis products used as driver data is given in Table 3. We use as basic driver data sea surface temperature (SST), sea surface salinity (SSS), chlorophyll a concentration (Chl a), mixed layer depth (MLD), bathymetry (BAT), distance from shore (DIST), ice concentration (ICE) and the change in ice concentration from the month to month (prior to current).

Chl a values during the dark winter season were set to 0. In addition to the reanalysis data, $p\mathrm{CO}_2$ values from the closest coastal grid cell of the open ocean map were used as a driver in our MLR. We neglect the approximately 1 $\mu$atm difference between partial pressure (reported in the mapped products) and fugacity of $\mathrm{CO}_2$ (reported in SOCAT) as it is much smaller than the accuracy of the data extracted from SOCAT v5 (2 to 10 $\mu$atm) and the uncertainty associated with the open ocean maps. The open ocean $p\mathrm{CO}_2$ values were extracted from two different products (Rödenbeck et al. (2014) (version oc_v1.5)

and Landschützer et al. (2017, 2016) (version 2016)). Rödenbeck et al. (2014) is based on a data-driven diagnostic model of mixed layer ocean biogeochemistry fitted against surface $p\mathrm{CO}_2$ observations while Landschützer et al. (2016) uses a two-step neural network (a feed-forward network coupled with self-organizing maps, FFN-SOM) trained with the $p\mathrm{CO}_2$ observations. Please note that the Rödenbeck open ocean map contains data in coastal grid boxes, while the Landschützer open ocean map is restricted to the open ocean regions. The MLR models based on these two are called MLR 1 (based on the coastal $p\mathrm{CO}_2$

values from the Rödenbeck map) and MLR 2 (based on the the nearest open ocean $p\mathrm{CO}_2$ values of the Landschützer map), respectively. To determine the extent to which the regressions benefit from the information in the open ocean maps, a third

**Table 3.** Products used as driver data in the MLR and the maps.

| Product used | Resolution | Reference |
| --- | --- | --- |
| Chl a for MLR | 4km x 4km, 8 days | Global Ocean Chlorophyll (Copernicus-GlobColour) |
| | | from Satellite Observations - Reprocessed |
| Chl a for maps | 4km x 4km, monthly | Global Ocean Chlorophyll (Copernicus-GlobColour) |
| | | from Satellite Observations - Reprocessed |
| MLD | 12.5km x 12.5km, monthly | Arctic Ocean Physics Reanalysis |
| ICE | $0.25°x0.25°$, monthly | Cavalieri et al. (1996) |
| SST / SSS | $0.25°x0.25°$, weekly | Global Ocean Observation-based Products |
| | | Global_Rep_Phy_001_021 |
| BAT | 2min x 2min | ETOPO2v2 Center (2006) |
| Rödenbeck $p$CO$_2$ | $5° \times 4°$, monthly | Rödenbeck et al. (2014) |
| Landschützer $p$CO$_2$ | $1° \times 1°$, monthly | Landschützer et al. (2017) |

MLR, MLR 3, was determined. Here, we do not use any of the open ocean maps as driver, but to account for the annual rise in $CO_2$, year is included in the set of driver data.

For preparing the input data for the MLR, observations closest to each SOCAT $f$CO$_2$ data point in time and space were extracted from the 3D fields with the driver data. This produces, for each of the driver data, a vector as long as the SOCAT $f$CO$_2$ observations. After this, the $f$CO$_2$ data as well as all extracted driver data were binned on a monthly $0.125°x0.125°$ grid covering 1997 to 2016. These procedures ensure that the driver data have the same bias in space and time within each grid box as the $f$CO$_2$ data. If a grid box for example only contains $f$CO$_2$ observations from the first week of the month and the northwestern corner, we make sure, that also the gridded driver data only contains values from the first week and the northwestern corner of the grid box, and not an average over the entire month and grid box. This is mostly important for the chlorophyll driver data, which are available at a very high resolution compared to the $f$CO$_2$ maps produced in this work. These driver data were used for determining the MLRs.

For producing the final maps, a second set of the driver data was prepared, in the following called field data. Here the driver data were directly regridded to a monthly $0.125°x0.125°$ grid, providing full spatial and temporal coverage and a homogeneous average in each grid box. The field data were used to produce the $f$CO$_2$ maps based on the equation derived MLR equations.

## 2.3 Multi linear regression

The multi linear regression models were constructed by forward and backward stepwise regression using the driver data as predictor variables to model the $f$CO$_2$ observations. In each step of this regression procedure, the model's tolerance to addition or exclusion of a variable is tested. This decision on whether to add or remove a term is based on the p-value of the F-statistic with or without the term in question. The entrance tolerance was set to $0.05$ and the exit tolerance to $0.1$. The model includes

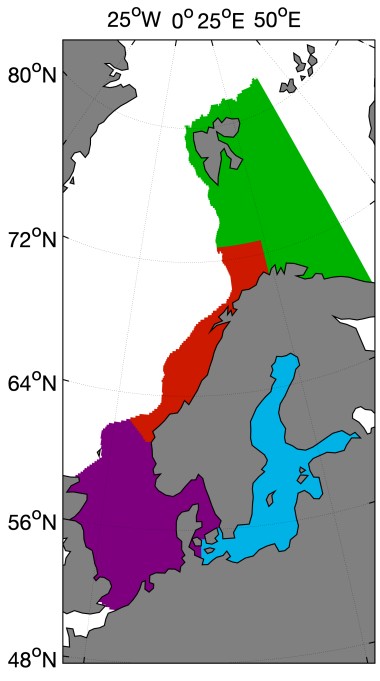

**Figure 1.** The study area and the location of the four different regions North Sea (purple), Norwegian Coast (red), Barents Sea (green) and Baltic Sea (blue).

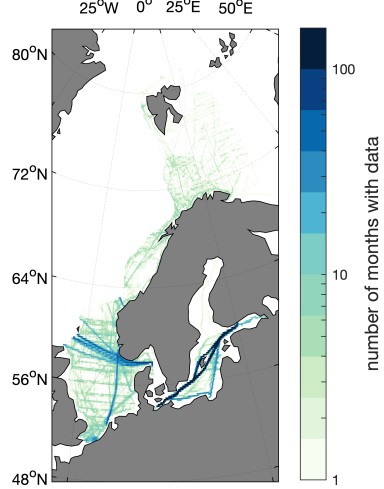

**Figure 2.** The number of months with $f$CO$_2$ data from SOCAT v5 in each grid box. The data cover a range of 20 years (240 months).

**Table 4.** Driver used in the different regressions.

| | log (MLD) | SST | SSS | CHL | ICE | ICE change | log (BAT) | DIST | $pCO_2$ | year |
|---|---|---|---|---|---|---|---|---|---|---|
| **North Sea** | | | | | | | | | | |
| MLR 1 | x | x | x | x | x | | x | | x | |
| MLR 2 | x | x | x | x | x | x | x | x | x | |
| MLR 3 | x | x | x | x | x | x | x | x | | x |
| **Norwegian Coast** | | | | | | | | | | |
| MLR 1 | x | x | x | x | x | | x | | x | |
| MLR 2 | x | x | x | x | x | x | | | x | |
| MLR 3 | x | x | x | x | x | x | x | | | x |
| **Barents Sea** | | | | | | | | | | |
| MLR 1 | x | x | x | x | | x | x | | x | |
| MLR 2 | x | x | x | x | x | x | | | x | |
| MLR 3 | x | x | x | x | x | x | | | | x |
| **Baltic Sea** | | | | | | | | | | |
| MLR 1 | x | x | x | x | x | x | x | | x | |
| MLR 2 | x | x | x | x | x | x | | | x | |
| MLR 3 | x | x | x | x | x | | x | | | x |

constant, linear, and quadratic terms as well as products of linear terms. Equation 1 gives the basic equation, with $X_1...X_n$ being the driver data and $a_1...a_{nn}$ the regression coefficients.

$$y = a_0 + a_1 \cdot X_1 + ... + a_n \cdot X_n + a_{12} \cdot X_1 X_2 + ... + a_{mn} \cdot X_m X_n + a_{11} \cdot X_1^2 + ... + a_{nn} \cdot X_n^2 \tag{1}$$

The $pCO_2$ value of the respective open ocean maps was used for MLR 1 and MLR 2, while year was always used as a driver variable in MLR 3. Inclusion of stationary drivers (such as month, latitude and longitude) in the MLR increased the performance of MLR 2 and MLR 3. However, these were still not better than MLR 1 and we therefore decided to limit this analysis to dynamic parameters. Using dynamic drivers only, assures a dynamic description of the conditions in the field, and gives us the possibility to reproduce changes caused by a regime shifts, for example the ongoing atlantification of the Barents Sea (Oziel et al., 2016; Lind et al., 2018).

## 2.4 Validation

The three linear fits were compared to each other in each region by taking into account the $R^2$ and the root mean square error (RMSE) of the fit, and the Nash Sutcliffe method efficiency (ME) (Nondal et al., 2009). The method efficiency compares how

well the model output ($E_\mathrm{n}$) fits the observations ($I_\mathrm{n}$) for every data point $n$ to how well a simple monthly average ($\bar{I}$) would fit the observations:

$$\mathrm{ME} = \frac{\sum_n \left(I_\mathrm{n} - E_\mathrm{n}\right)^2}{\sum_n \left(I_\mathrm{n} - \bar{\mathrm{I}}\right)^2} \tag{2}$$

A method efficiency >1 means that using just monthly averages of all data in the region would fit better to measured data than the respective model. Generally, a method efficiency >0.8 is considered bad. Besides the statistics of the fit itself, the final maps were also compared to the gridded SOCAT v5 data, resulting in an average offset and standard deviation. In order to compare the maps against data that were not used to produce the maps, we predicted the $f\mathrm{CO}_2$ for the years 2017 and 2018 (i.e., we applied the trained multi-linear model to driver data from 2017 and 2018) and compared these maps to $f\mathrm{CO}_2$ observations in SOCAT v2019, gridded on a monthly 0.125°x0.125° grid. We also compare the maps directly with observations from repeated sampling locations in the North Sea and the Baltic Sea.

## 2.5 Ocean acidification

For calculating the pH, alkalinity (AT) was estimated in the North Sea, along the Norwegian Coast, and in the Barents Sea via a salinity-alkalinity correlation after Nondal et al. (2009). Alkalinity describes the capacity of the sea water to buffer changes in pH. As the concentration of most of the weak bases in seawater is strongly dependent on the salinity, alkalinity can in many regions be estimated from salinity. However, in regions with a high amount of organic bases in seawater, for example in strong blooms or at river mouths, deviations from the alkalinity-salinity relationship can occur. The carbonate system was calculated using the CO2SYS program (van Heuven et al., 2009) with carbonic acid dissociation constants of Mehrbach et al. (1973) as refitted by Dickson and Millero (1987), $\mathrm{KSO}_4^-$ dissociation constants after Dickson (1990) and the boron-salinity relation after Uppström (1974). For the Baltic Sea, we did not calculate pH as the alkalinity-salinity relationship in this region is complex due to different AT-S relations in different sub-regions of the Baltic Sea, and a non-negligible increase of AT over the last 25 years (Müller et al., 2016).

## 2.6 Calculation of trends

For calculating trends of $f\mathrm{CO}_2$ and ocean acidification, the data in every grid box were deseasonalised by subtracting the long-term averages of the respective months. Then a linear fit was applied to the deseasonalised time-series. For illustrating the influence of interannual variability we calculated the trend for different time ranges. As a time range less than 10 years barely resulted in significant trends, we decided to limit the trend analysis to starting years 1998 through 2006 and ending years 2008 through 2016.

## 2.7 Flux calculation

The air-sea disequilibrium was calculated as the difference between our mapped $f\mathrm{CO}_2$ values and atmospheric $f\mathrm{CO}_2$ in each grid cell and time step. The atmospheric $f\mathrm{CO}_2$ was determined by converting the $x\mathrm{CO}_2$ from the NOAA Marine Boundary

Layer Reference product from the NOAA GMD Carbon Cycle Group into $f\mathrm{CO}_2$ by using monthly SST and SSS data (Table 3) and monthly air pressure data from the NCEP-DOE Reanalysis 2 (Kanamitsu et al., 2002). We calculated the air-sea $\mathrm{CO}_2$ flux (F) according to Equation 3, such that negative fluxes are into the ocean. The gas transfer coefficient $k$ was determined using the quadratic wind speed (u) dependency of Wanninkhof (2014) (Equation 4). The Schmidt number, $Sc$, was calculated according to Wanninkhof (2014) and the solubility coefficient for $\mathrm{CO}_2$, $K_0$, after Weiss (1974).

$$\mathrm{F} = k \cdot K_0 \cdot (f\mathrm{CO}_{2,\mathrm{sw}} - f\mathrm{CO}_{2,\mathrm{atm}}) \tag{3}$$

$$k = a_q \cdot \langle u^2 \rangle \cdot \left( \frac{Sc}{660} \right)^{-0.5} \tag{4}$$

In our calculations, we used 6-hourly winds of the NCEP-DOE Reanalysis 2 product. The coefficient $a_q$ in Equation 4 is strongly dependent on the used wind product (Roobaert et al., 2018). We determined it to be $a_q = 0.16$ cm h$^{-1}$ for the 6-hourly NCEP 2 product following the recommendations of Naegler (2009) and by using the World Ocean Atlas sea surface temperatures (Locarnini et al., 2018). The barrier effect of sea ice on the flux was taken into account by relating the flux to the ice cover extent following Loose et al. (2009). As the gas exchange in areas that are considered 100% ice covered from satellite images should not be completely neglected, we use a sea ice barrier effect for a 99% sea ice cover in all grid cells where the sea ice coverage exceeded 99%.

## 3 Results

### 3.1 Maps of $f\mathrm{CO}_2$

The skill assessment metrics for MLR 1, MLR 2 and MLR 3 are presented in Table 5. It shows the the $R^2$ and RMSE of the fit, the ME, as well as the average offset and standard deviation to the gridded SOCAT data. The coefficients for MLR 1, MLR 2 and MLR 3 are provided in the supplement. The MLRs substantially improve the predictions of the open ocean maps in all studied regions, showing a better average offset and standard deviation to SOCAT v5 and ME than the coarser-resolution open ocean maps (for example: Rödenbeck map: North Sea $0\pm95$ $\mu$atm, Norwegian Coast: $2\pm17$ $\mu$atm, Barents Sea: $22\pm40$ $\mu$atm, Baltic Sea: $4\pm48$ $\mu$atm; MLR1: North Sea: $0\pm26$ $\mu$atm, Norwegian Coast: $0\pm16$ $\mu$atm, Barents Sea: $0\pm19$ $\mu$atm, Baltic Sea: $2\pm42$ $\mu$atm). In all regions MLR 1 has the best model efficiency, the highest $R^2$ and the smallest RMSE of the fit, while these statistics are worse for MLR 2 and MLR 3. This can be explained by the fact that the Rödenbeck map contains information about the continental shelf and the Barents Sea, while for MLR 2 the closest open ocean grid cell of Landschützer et al. (2017) was used. The fact that MLR 3 showed the weakest performance shows the value of using information from the open ocean maps in the regression.

Figure 3 shows, from left to right, the spatial distribution of the average difference between the predicted $f\mathrm{CO}_2$ by MLR 1 and the gridded SOCAT v5 data, the Rödenbeck map and the gridded SOCAT v5 data, the difference between MLR 1 and the Rödenbeck map, and, for comparison, between MLR 3 and the SOCAT v5 data. In the North Sea, MLR 1 seems

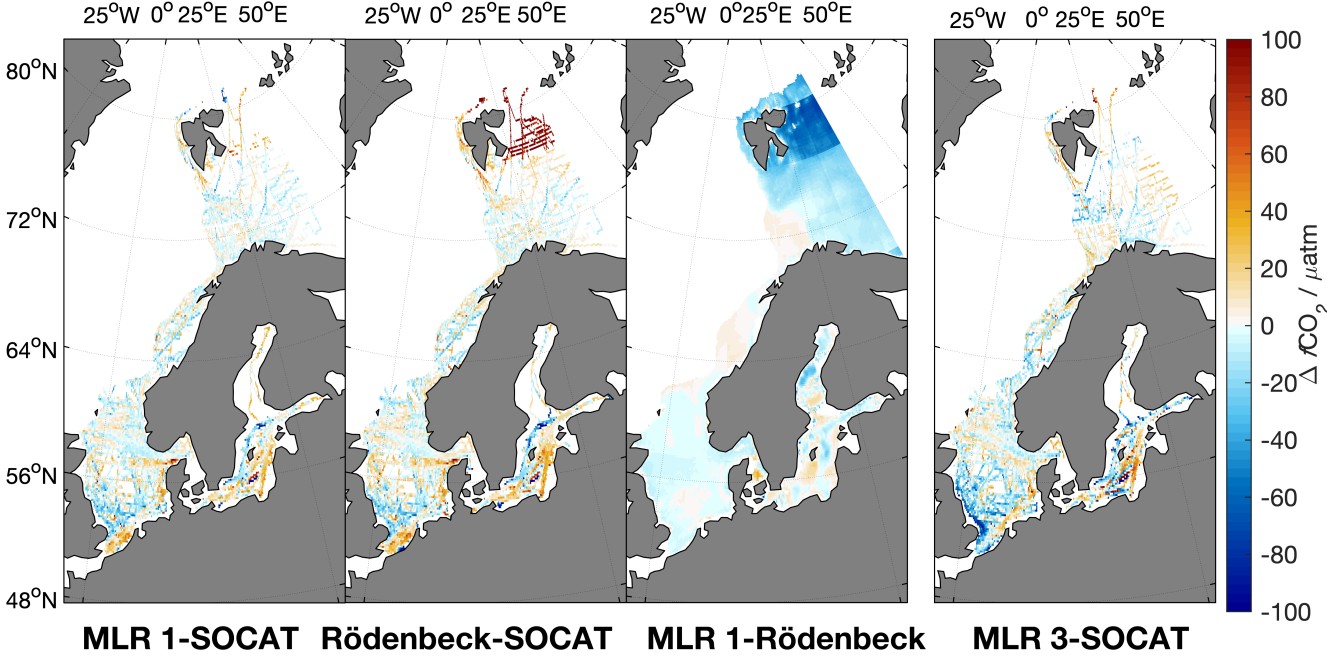

**Figure 3.** Average regional differences between MLR 1 and gridded SOCAT v5 data, the Rödenbeck map and gridded SOCAT v5 data, MLR 1 and the Rödenbeck map, and MLR 3 and the gridded SOCAT v5 data (from left to right).

to slightly overestimate the $f\mathrm{CO}_2$ in the constantly mixed region at the entrance of the English channel and the area off the Danish North Sea coast. In the Baltic, MLR 1 generally describes the spatial variability in $f\mathrm{CO}_2$ well. However, in the Gulf of Finland it usually predicts too low $f\mathrm{CO}_2$ values during May/June while it slightly underestimates events of very high $f\mathrm{CO}_2$ in December/January. Regardless, the spatial biases vs SOCAT are clearly smaller for MLR 1 than for the original Rödenbeck

map. Further, as the predictions of MLR 2 and 3 are clearly inferior to those of MLR 1 (Table 5 and Figure 3 (MLR 3 only)), we will use MLR 1 results for the further analyses. An extended validation of the MLR 1 maps can be found in the discussion section.

Figure 4 shows the monthly averages of $f\mathrm{CO}_2$ produced by MLR 1 for February, May, August and November. In all regions, the highest $f\mathrm{CO}_2$ values occur in the winter, while the lowest $f\mathrm{CO}_2$ occur in summer. The largest seasonal cycle could be

observed in the Baltic Sea, where $f\mathrm{CO}_2$ reached well below 200 $\mu$atm in mid summer and over 500 $\mu$atm during the winter.

We notice that the gradients that exist between the grid cells in the Rödenbeck map, are still visible in our maps in some regions, for example the sharp gradient in the southern North Sea in February, or the east-west and north-south gradients in the entire North Sea in August. Such gradients are also evident in directly mapped $p\mathrm{CO}_2$ data (Kitidis et al., 2019), however, here they are strongly meridional and latitudinal in their extent. As such, while these gradients do reflect actual features of the $f\mathrm{CO}_2$

distribution in the North Sea, their specific shape here, are also a consequence of the influence of the Rödenbeck maps on our

**Table 5.** Statistical evaluation of the MLR 1, MLR 2 and MLR 3 in comparison to the open ocean maps of Rödenbeck et al. (2015) and Landschützer et al. (2017) for each region. The data for the open ocean map of Landschützer et al. (2017) are in parentheses since this is based on an extrapolation of the nearest open ocean grid cell towards the coast. The number of grid cells containing data is given behind the region abbreviations.

| | $R^2$ adj | RMSE | ME median | difference to gridded SOCAT v5 | |
| | | | | mean | standard deviation |
| | | /$\mu$atm | | /$\mu$atm | /$\mu$atm |
|---|---|---|---|---|---|
| **North Sea** (36170) | | | | | |
| MLR 1 | 0.7271 | 25 | 0.3145 | -0.15 | 26 |
| MLR 2 | 0.5130 | 33 | 0.5789 | -0.52 | 36 |
| MLR 3 | 0.5331 | 33 | 0.4895 | -2.4 | 32 |
| Rödenbeck | | | 0.3522 | -0.28 | 95 |
| (Landschützer) | | | 0.5714 | -4.7 | 103 |
| | | | | | |
| **Norwegian Coast** (16014) | | | | | |
| MLR 1 | 0.7860 | 16 | 0.1742 | 0.46 | 16 |
| MLR 2 | 0.5634 | 22 | 0.3597 | -2.3 | 24 |
| MLR 3 | 0.6074 | 20 | 0.2436 | -0.08 | 21 |
| Rödenbeck | | | 0.2177 | 2.0 | 17 |
| (Landschützer) | | | 0.3294 | 7.0 | 23 |
| | | | | | |
| **Barents Sea** (13925) | | | | | |
| MLR 1 | 0.8871 | 12 | 0.1069 | 0.32 | 19 |
| MLR 2 | 0.8724 | 14 | 0.0986 | 1.3 | 68 |
| MLR 3 | 0.8672 | 18 | 0.1082 | 1.3 | 24 |
| Rödenbeck | | | 0.2923 | 22 | 40 |
| (Landschützer) | | | 0.3364 | 15 | 44 |
| | | | | | |
| **Baltic Sea** (46810) | | | | | |
| MLR 1 | 0.9076 | 39 | 0.0488 | 2.2 | 42 |
| MLR 2 | 0.6733 | 66 | 0.3111 | -1.0 | 68 |
| MLR 3 | 0.6628 | 67 | 0.3027 | 0.24 | 69 |
| Rödenbeck | | | 0.1326 | 4.2 | 48 |

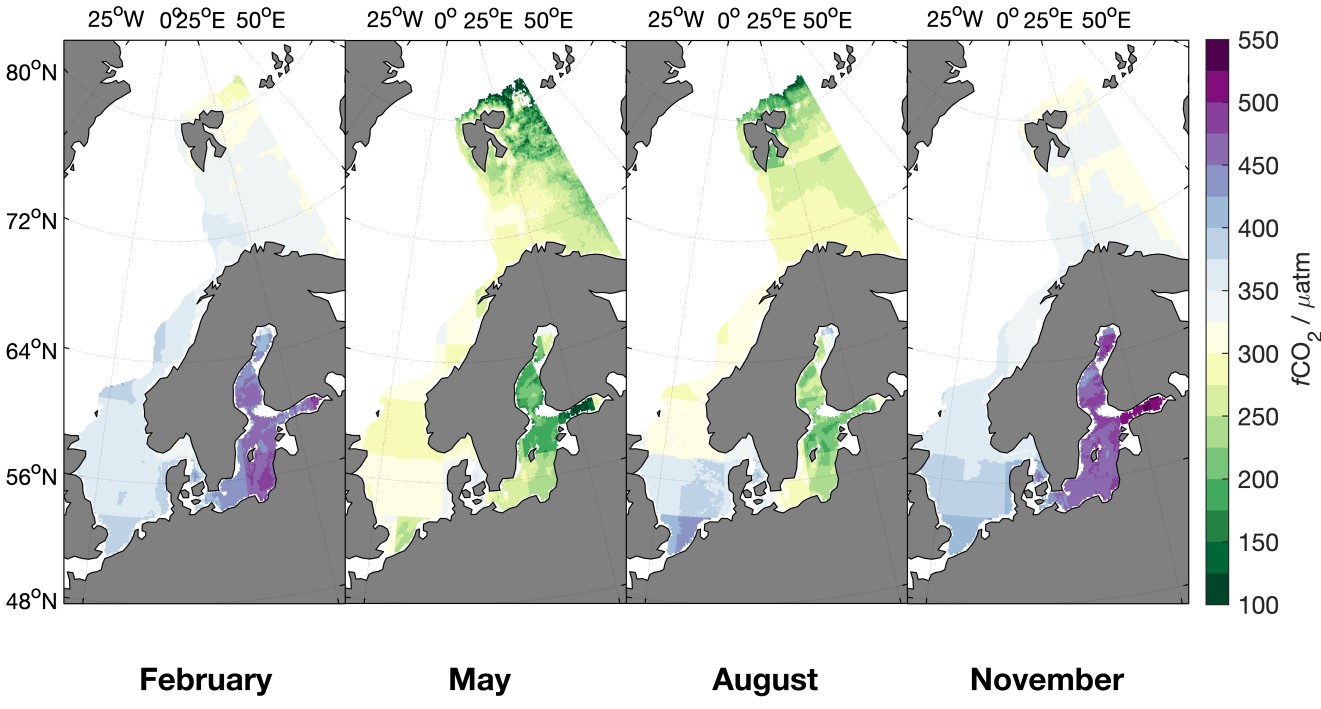

**Figure 4.** The average $f\text{CO}_2$ of MLR 1 (1998-2016) for one example months in each season (February, May, August and November).

estimates; from the use of these maps as a driver in the MLR and their importance in improving the statistical performance vs the MLR that did not use these values as a driver (MLR 1 vs MLR 3, Table 5). Also, they do reflect the uncertainty of - and our level of confidence in - the estimated $p\text{CO}_2$ values; being approximately similar to or slightly larger than the RMSE of MLR 1 (Table 5). Any smoothing would be completely artificial, and, while being more visually pleasing, would not better reflect the
5  truth in any meaningfully quantifiable extent. We have therefore chosen to leave them untouched. These gradients are therefore also visible in subsequent pH and trend maps.

## 3.2 Maps of pH

The monthly average of pH calculated from MLR 1 $f\text{CO}_2$ ranges from about 8 during winter to 8.15 during summer in the North Sea and at the Norwegian coast (Figure 5). Towards the Barents Sea the pH maximum during summer increases to 8.2.
10  The pH of 8.00 - 8.15 in regions with a large influence from the Atlantic, such as the northern North Sea and the Norwegian coast, is in good agreement with the range of pH determined for the open North Atlantic (Lauvset and Gruber, 2014; Lauvset

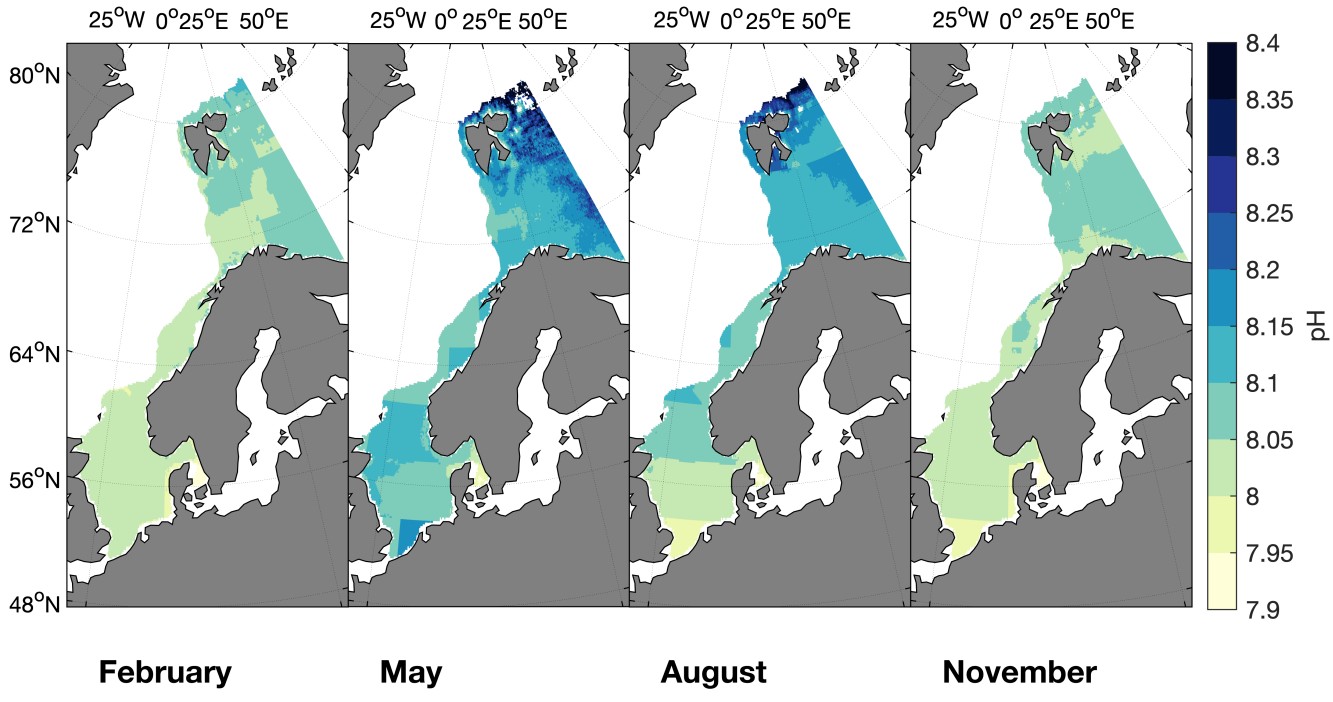

**Figure 5.** The average pH based on MLR 1 (1998-2016) for one example month in each season (February, May, August and November).

et al., 2015). In the North Sea, the pH is in the same range as reported in Salt et al. (2013) and it also shows the same distribution in August/September, with higher pH in the northern North Sea and lower pH in the southern part.

## 4 Discussion

### 4.1 Performance of the $p$CO$_2$ maps

The performance of the MLR and the maps are evaluated in different ways: (1) the R$^2$ and the RMSE of the fit, (2) the average deviation and its standard deviation, as well as the ME between the produced $f$CO$_2$ maps and the gridded observations as a regional average, (3) showing the median deviation between the MLR and the gridded observations on a monthly level, and (4) by comparing the data from the $f$CO$_2$ maps to observations from two time series stations. (2) - (4) will be shown for the time period covered by the driver data (1998-2016) and for the prediction of the $f$CO$_2$ values for 2017 and 2018. These predicted values are compared with data from the newest SOCAT release (SOCATv2019) and provide a comparison with an independent

dataset. Please note that the comparability of the model performance between the different regions is limited. All used statistical parameters are influenced by characteristics that can vary substantially between the different regions, such as range of the data, their variability or the amount of grid cells with data. For example, in a diverse region with many measurements the amount of variability captured by these measurements is most likely larger and, thus will lead to a weaker correlation.

Generally, the uncertainty of MLR 1 is in the same range as in other studies (Laruelle et al., 2017; Yasunaka et al., 2018) mapping coastal $f\mathrm{CO}_2$ dynamics: 25 $\mu$atm in the North Sea, 16 $\mu$atm along the Norwegian Coast, 12 $\mu$atm in the Barents Sea, and 39 $\mu$atm in the Baltic Sea (based on the RMSE in Table 5). In the Baltic Sea, which has a large variability in itself, Parard et al. (2016) obtained lower standard deviations through dividing the area in smaller sub-regions.

One clear drawback of the here presented MLR 1 is the clearly visible grid-pattern of the open ocean $p\mathrm{CO}_2$ product that was
used as input data with its grid size of 5 x 4 $^\circ$, mentioned in Sect 3.1. There are two ways how one could get rid of this artifact in a future release. A finer resolution of the used open ocean maps will lead to a better representation of the actual gradients in our mapped product. Rödenbeck et al. just released a newer, finer resolution of their open ocean product (2.5 x 2$^\circ$) that we intend to use in a future version of this data product. However, this will not be sufficient to eradicate the artifact completely. Another approach, running the MLR without an open ocean $p\mathrm{CO}_2$ product can provide a coastal $p\mathrm{CO}_2$ product without this
artifact. While in principle it is preferential to have coastal maps that are independent of the open ocean products, MLR 3, which is running without open ocean $p\mathrm{CO}_2$ as driver, did clearly not reach the same accuracy as MLR 1 (Table 5). New and better input fields or a different regression method could help improving the independent coastal maps in the future. Another impact that the open ocean $p\mathrm{CO}_2$ product of Rödenbeck et al can have on MLR 1 is the potential introduction of patterns from regions further away as the spatial correlations used in producing the Rödenbeck et al. $p\mathrm{CO}_2$ just ignore land barriers. However,
the influence of these spatial correlations is relatively small in regions with a high data density (as the European shelf) and the multi linear regression used to produce MLR 1 corrects for this.

The seasonal differences between MLR 1 determined values and the SOCAT v5 data for each region are shown in Figure 6. This comparison shows a very good agreement. For MLR 1, the seasonal variations of the median bias are small in the North Sea, along the Norwegian coast and in the Baltic Sea. In the Barents Sea, however, the bias varies seasonally. Here, MLR 1
slightly underestimates the $f\mathrm{CO}_2$ in winter and early spring, while it overestimates the $f\mathrm{CO}_2$ in summer. In all other regions, the median seasonal bias is smaller than the uncertainty of the maps. The larger seasonal bias in the Barents sea is most likely caused by the larger seasonal bias in the number of available observations. There are no data available in October, December and January.

When comparing all observations from the years 2017 and 2018 to the predictions by the MLR 1, we find a good agreement
in the North Sea ($2 \pm 20$ $\mu$atm) and no seasonal bias (Figure 7). In the other regions, the agreement is somewhat reduced compared to the years 1997-2016 ($-9 \pm 39$ $\mu$atm (Norwegian Coast), $-5 \pm 29$ $\mu$atm (Barents Sea) and $28 \pm 58$ $\mu$atm (Baltic Sea)). In these regions we also observe a seasonal bias in the years 2017 and 2018. At least for the Baltic Sea this could be a result of the extraordinary warm and dry summer in 2018, that led to very low $f\mathrm{CO}_2$ values (see Figure 8 and the data in SOCAT (Bakker et al., 2016)). Please note, that for this comparison the MLR was extrapolated in time. Only observations until
December 2016 were used to produce the MLR. Therefore accuracy of the maps itself is reduced.

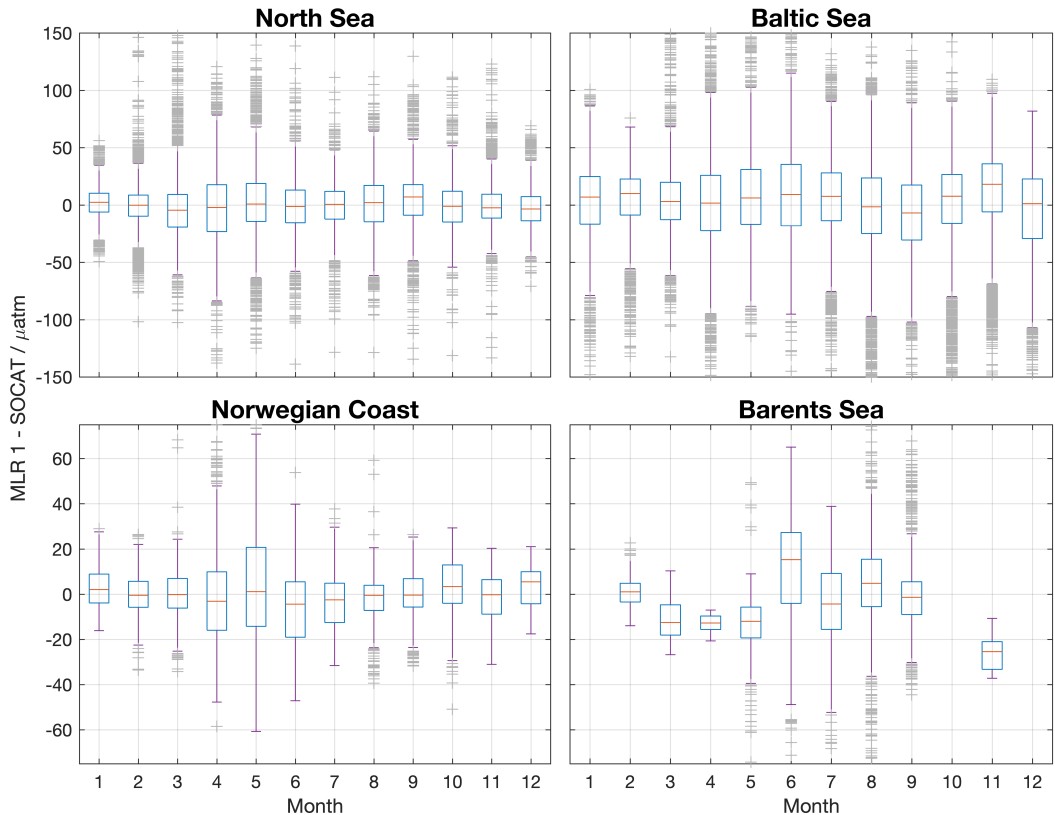

**Figure 6.** Boxplots showing the median deviation of MLR 1 from the gridded SOCAT 5 data for each region (red line). The boxes show the upper and lower $75\%$ percentiles. $99\%$ of the data lays within the range of the purple whiskers. Extremes are shown as gray crosses.

In a second test to investigate to which extent MLR 1 can reproduce observations we compared the MLR output with time series data from two voluntary observing ship lines in two very different regions with a good data coverage: M/V Nuka Arctica in the northern North Sea ($0$-$2°$E, $58$-$60°$N) and M/V Finnmaid in the Baltic Sea ($23$-$24°$E, $59$-$60°$N) (Figure 8). The agreement between the MLR 1 and the observations is very good. MLR 1 reproduces the general seasonality and some of the interannual variability, also in the years 2017 and 2018, of which the observations were not used in the regression.

When performing interpolation exercises it is always important to be aware of the fact that the resulting maps might come with biases and do not represent all regions equally well. While the here presented maps give a good general overview about the surface ocean $f\mathrm{CO}_2$ variability in regions with a relatively large amount of data, the reliability, however, is limited in regions where the data coverage is more scarce. This is especially the case, when the region with scarce data coverage is showing different characteristics in, for example, temperature and salinity, compared to the rest of the region. One such example is the Gulf of Bothnia in the Baltic Sea region where almost all data used to derive the MLR is from south of $60°$N i.e. not in the Gulf of Bothnia, but in the Baltic Proper and western Baltic Sea (see Figure 2). The MLR method can also lead to unrealistic

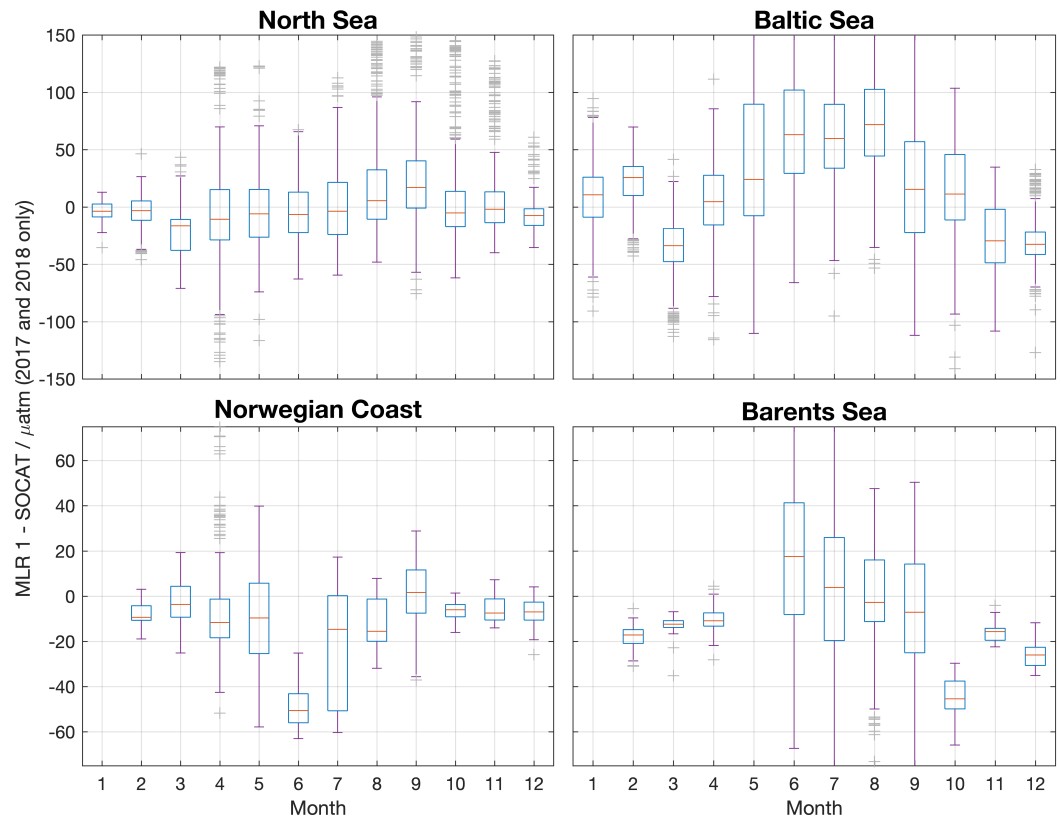

**Figure 7.** Boxplots showing the median deviation between MLR 1 (based on observations until 2016) and measured $f\mathrm{CO}_2$ values in 2017 and 2018. The boxes show the upper and lower $75\%$ percentiles. $99\%$ of the data lays within the range of the purple whiskers. Extremes are shown as gray crosses. The number of grid cells with data available were: North Sea: 5047, Norwegian Coast: 1543, Barents Sea: 2312, Baltic Sea: 5414.

extreme values and even negative $f\mathrm{CO}_2$. Some such values occur in the northeastern Barents sea as well as in parts of the Baltic Sea (about $0.01\%$ of the grid cells in each region). As pH cannot be calculated for negative $f\mathrm{CO}_2$, we excluded all negative $f\mathrm{CO}_2$ values for the calculation of pH. Excluding the negative values resulted in a change of the average $f\mathrm{CO}_2$ of 0.05 $\mu$atm (Baltic Sea) and 0.3 $\mu$atm (Barents Sea) so they are of negligible importance for the flux estimates. While the negative values
5 are easy to spot and discard there are most likely other unrealistically low values in spring and summer in the very north and northeastern Barents Sea as well as some parts of the Baltic Sea. However, there are no data available in SOCAT v5 or elsewhere available to validate this.

All regions with questionable $f\mathrm{CO}_2$ are also questionable in their pH data. There is a number of very high pH in the Barents Sea (Figure 5), that are associated with also very low $f\mathrm{CO}_2$ (4) that might not be realistic. In addition, estimated pH values

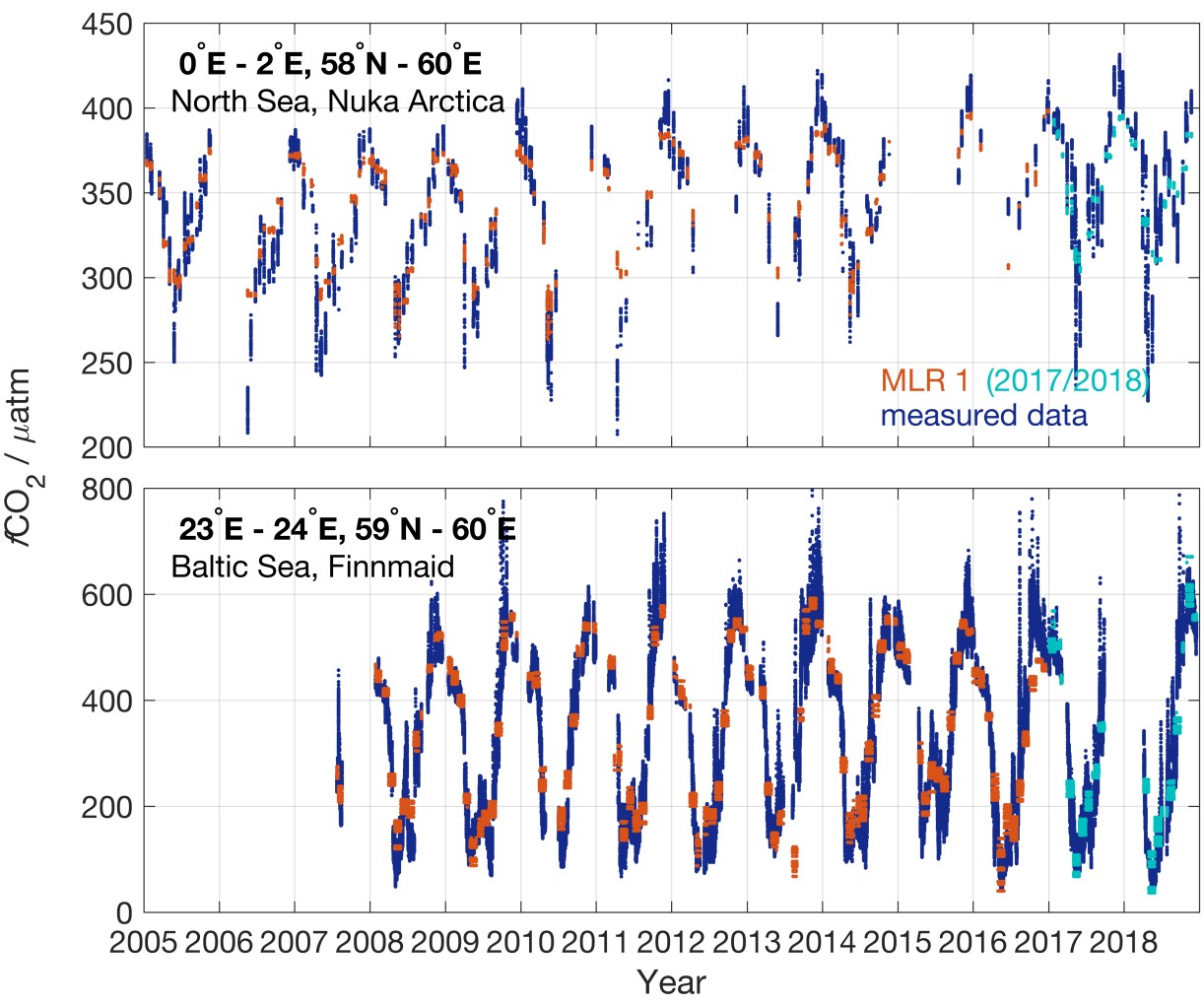

**Figure 8.** Time series of VOS data from Nuka Arctica (upper panel, blue) and Finnmaid (lower panel, blue) compared with MLR 1 at the same location (red). In light blue the predictive MLR output for the years 2017 and 2018 is shown.

in low salinity regions where the actual alkalinity-salinity deviates strongly from the Nondal et al. (2009) one used here (e.g. river mouths in the southern North Sea or the Skagerrak), should be interpreted with caution.

## 4.2 Trends in $f\mathrm{CO_2}$ and pH

Trends in surface ocean $f\mathrm{CO_2}$ in coastal regions are often difficult to assess because of the scarcity of data relative to the highly dynamical character of these regimes and their large interannual variability. For example, the start of the productive season can range from February to April even within a small area, such that even restricting the analysis to specific seasons (e.g. winter) can be challenging. Also, due to lack of data, especially winter data, most observational studies are based on summer data. Further, the fact that these measurements typically do not take place every year, adds even more uncertainty to the estimated trend, as interannual variability can mask the trend signal.

The monthly maps of $f\mathrm{CO_2}$ from 1998 to 2016 enable us now to estimate the trend in surface ocean $f\mathrm{CO_2}$ for the entire region, equally distributed over the seasons (Figure 9, left). All trends were computed from deseasonalized data. The interannual variability of the trend estimates in each region is shown in the panels on the right hand side in Figure 9. We exclude the northern Baltic Sea from the trend map because we do not expect to have a robust trend estimate in that region as there are only very few data from that region in the regression. Based on the linear regression the significant trends in $f\mathrm{CO_2}$ have an average uncertainty of 0.5 $\mu$atm/yr (North Sea), 0.4 $\mu$atm/yr (Norwegian Coast), 0.4 $\mu$atm/yr (Barents Sea), and 0.7 $\mu$atm/yr (Baltic Sea), while the shorter time periods shown have a higher, no longer time periods than 1998-2016 (for which the given uncertainties of the trend apply) are shown. For pH trends, the average uncertainties of the regressions over 1998-2016 are $5 \cdot 10^{-4}$ (North Sea) and $7 \cdot 10^{-4}$ (Norwegian Coast and Barents Sea).

In most of the regions addressed in this study, the trend in the surface ocean is lower than the trend in atmospheric $x\mathrm{CO_2}$ (global average 2.02 ppm yr$^{-1}$ ("Cooperative Global Atmospheric Data Integration Project", 2015)). Trends exceeding the atmospheric values in the period from 1998 to 2016 can only be observed at the entrance of the English Channel, in Storfjorden/Svalbard and the Gulf of Finland ($2.5 - 3$ $\mu$atm yr$^{-1}$). It has to be noted that there was almost no measured $f\mathrm{CO_2}$ as MLR input from Storfjorden. Therefore, these trends are highly uncertain. The trend in the western North Sea is only slightly lower than the trend in the atmosphere ($1.5 - 2$ $\mu$atm yr$^{-1}$), while the trends in the eastern North Sea, along the Norwegian coast and in the Barents Sea are lower (0.5-1.5 $\mu$atm yr$^{-1}$). In the North Sea this is consistent with a recent study of Omar et al. (2019), which is directly based on observations. These low trends will result in an increase in the strength of the ocean carbon sink with time.

Generally, only few regressions over time ranges of less than 10 years turned out to be significant. This is an important finding when comparing the trends determined from our maps with the trends reported in literature, of which many were covering periods shorter than 10 years (Table 1). In order to compare the general patterns of $f\mathrm{CO_2}$ trends determined from our maps with those directly determined from observations over a similar time range, we estimated the $f\mathrm{CO_2}$ trends also from the SOCAT v5 observations that were used to produce the MLR (Table 6). We gridded and deseasonalized the SOCAT v5 data and divided the entire region into 9 subregions. A figure showing the fits and the data coverage can be found in Appendix A. These observation based trends show similar general patterns as those based on our maps (Figure 9, 1998-2016): (1) largest trends in

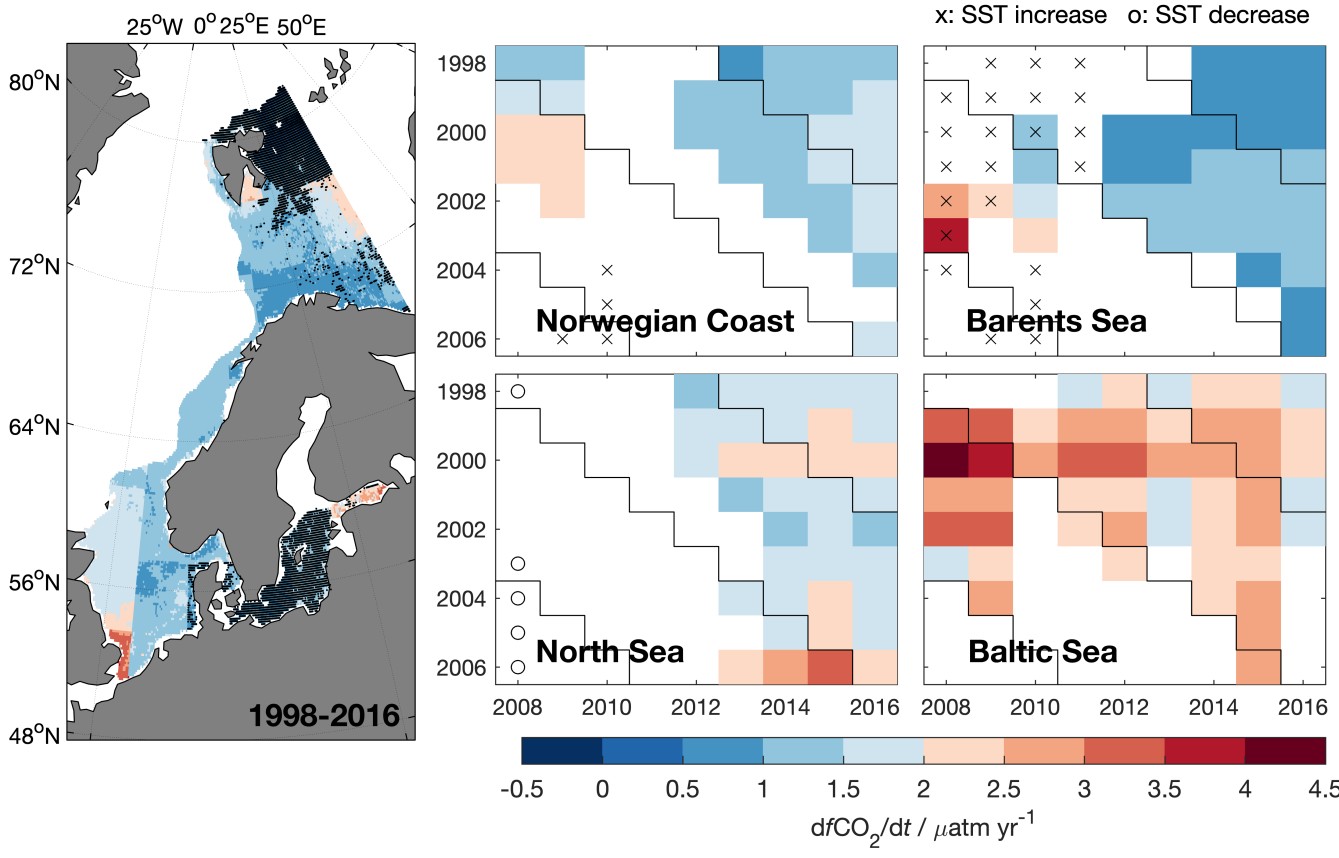

**Figure 9.** The trend in surface ocean $f\mathrm{CO_2}$ estimated from deseasonalized $f\mathrm{CO_2}$. The left hand panel show the spatial distribution of the trend over the time period from 1998 to 2016. Grid boxes without a significant trend are denoted with a black dot. The four right hand panels show the trends in different time periods in four regions, from the various years on the y-axis to the various years on the x-axis. Non significant trends were left blank. Significant trends in sea surface temperature are indicated with crosses/circles. The colorbar is centered on the approximate annual $f\mathrm{CO_2}$ rise in the atmosphere (2 $\mu$atm/yr)

**Table 6.** $f\mathrm{CO}_2$ trend calculated from gridded, deseasonalized SOCAT v5 observations.

| Region | Latitude / $^\circ$N | Trend / $\mu$atm yr$^{-1}$ |
| --- | --- | --- |
| North Sea, South | 51 - 54.5 | $3.2 \pm 1.3$ |
| North Sea, Center | 54.5 - 58 | $1.43 \pm 0.21$ |
| North Sea, North | 58 - 62 | $2.320 \pm 0.089$ |
| | | |
| Norwegian Coast, South | 62 - 68 | $2.12 \pm 0.19$ |
| Norwegian Coast, North | 68 - 73 | $1.426 \pm 0.099$ |
| | | |
| Barents Sea, South | 69 - 74 | $1.31 \pm 0.30$ |
| Barents Sea, North | 74 - 85 | $1.01 \pm 0.22$ |
| | | |
| Baltic Sea, South | 54 - 56 | $2.05 \pm 0.12$ |
| Baltic Sea, North | 56 - 61 | $1.84 \pm 0.21$ |

the southern North Sea, (2) decreasing towards the north with trends around the atmospheric trend in the northern North Sea and trends around 1 $\mu$atm yr$^{-1}$ in the Barents Sea, (3) close to atmospheric trends in the Baltic Sea.

The observation that some subareas (the Baltic Sea or along the shore of the western North Sea) did not show a significant trend can be explained by the fact that coastal sea systems, especially enclosed areas as the Baltic Sea, experience a high
anthropogenic pressure. Anthropogenic impacts other than rising atmospheric $\mathrm{CO}_2$ concentrations influence the ocean carbon system, for example the nutrient load of rivers can effect coastal ecosystems and primary production through eutrophication. This will result in lower $f\mathrm{CO}_2$ in summer and higher $f\mathrm{CO}_2$ in winter (Borges and Gypens, 2010; Cai et al., 2011). Another important process that influences the carbon system in the Baltic Sea are inflow events from the North Sea. In between such events, $\mathrm{CO}_2$ accumulates in deeper water layers causing an increasing gradient of dissolved inorganic carbon (DIC) across the
halocline. Whenever deep winter mixing occurs, this will then lead to a large increase of surface $f\mathrm{CO}_2$ because of the input of DIC rich waters from below. Another reason is the observed change in alkalinity with time. This affects the $f\mathrm{CO}_2$ though changes in the buffer capacity of the inorganic carbon system (Müller et al., 2016).

In most other regions, the sea surface $f\mathrm{CO}_2$ trends were typically smaller than the rise in the atmospheric $\mathrm{CO}_2$ concentration. A possible explanation is an earlier onset of the spring bloom. For example, in the North Sea a significant drawdown in $f\mathrm{CO}_2$
has been observed as early as February in some years, but there is also a large variability (Omar et al., 2019). The bloom timing and onset in the North Sea after the 1990s has been shown to be mainly triggered by the spring-neap tidal cycle and the air temperature (Sharples et al., 2006). The bloom timing and onset was found to be significantly earlier in the 2010s compared to the previous decades (Desmit et al., 2020). Even if the trend in winter $f\mathrm{CO}_2$ was following the atmospheric $x\mathrm{CO}_2$ increase, such a change in bloom timing and onset would lead to a trend lower than in the atmosphere when averaging over the entire
20  year. Figure 10a shows the annual trends in $f\mathrm{CO}_2$ in each month in the four regions considered. Particularly in the North Sea

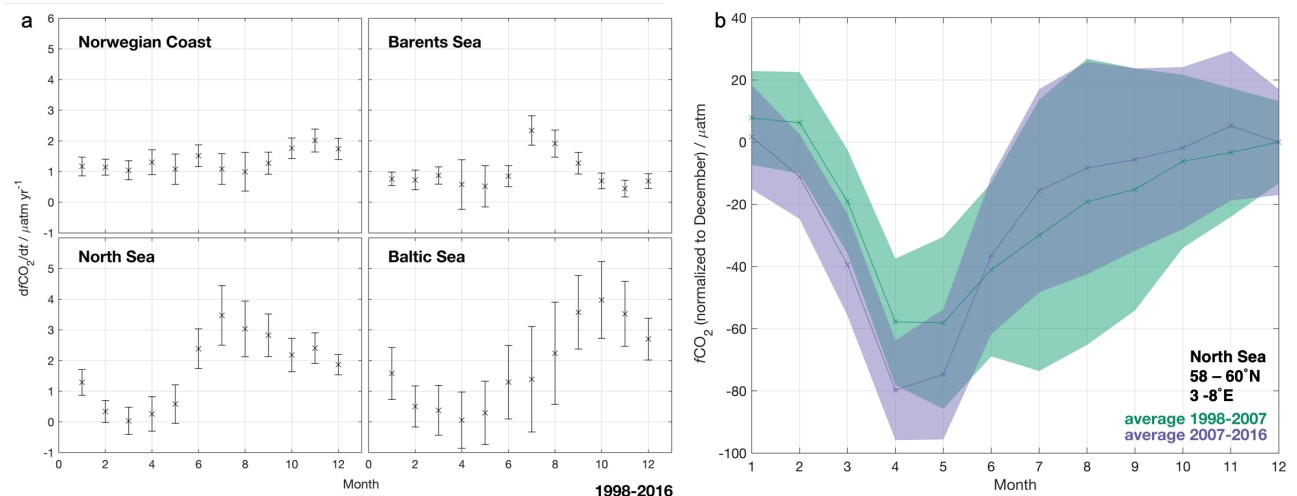

**Figure 10.** (a)The long term trend (1998-2016) in surface ocean $f\mathrm{CO}_2$ each month. (b) The average seasonality in $f\mathrm{CO}_2$ for the periods 1998-2007 (green) and 2007-2016 (purple) in the northeastern North Sea (58 - 60°N, 3 -8°E), normalized to December. The standard deviation for each month is shown as shaded area.

and Baltic Sea, very low $f\mathrm{CO}_2$ trends are observed in February – May, suggesting that changing timing of the spring bloom might be important here. Investigating the seasonal $f\mathrm{CO}_2$ in more detail (Figure 10b), revealed an earlier and deeper $f\mathrm{CO}_2$ drawdown in the second decade of our analysis (2007-2016) than in the first (1998-2007) in the northeastern North Sea (58 – 60°N, 3 -8°E). This strongly suggest that an earlier and stronger spring bloom is lowering the annual $f\mathrm{CO}_2$ growth rates in

this region, which is among the ones with the smallest $f\mathrm{CO}_2$ trends in the North Sea (about 1 $\mu$atm yr$^{-1}$, Fig. 9). In the other regions, no such changes could be established with confidence. Future investigations should aim at generating $f\mathrm{CO}_2$ maps with higher temporal resolution, as changes in the timing of the spring bloom might be a matter of days or weeks, which would not be fully resolved by the monthly maps presented here.

  When looking at the interannual variability, it becomes obvious that the trend in the North Sea is slightly smaller than the

atmospheric $\mathrm{CO}_2$ trend. In contrast, the Norwegian coast and the Barents Sea experience a robust trend much lower than the atmospheric trend (Norwegian Coast: $1 - 1.5$ $\mu$atm yr$^{-1}$, Barents Sea: around 1 $\mu$atm yr$^{-1}$). Here we can also see a stable pattern of warming over time scales of 10 to 15 years. The warming in itself would result in an increase of $f\mathrm{CO}_2$ with time, in addition to the atmospheric forcing. As we are observing a trend smaller than the atmospheric trend, temperature effects can't be the driver here. The lower trend stems most likely from an earlier onset of spring bloom. It has been shown that the

atlantification and the reduced ice coverage of the Barents sea leads to a longer productive season, and this will result in more months with strong undersaturation in $\mathrm{CO}_2$ (Oziel et al., 2016). In the Baltic Sea the pattens are different. Here the variability is much larger, while most of the time periods show a trend larger than the atmospheric trend ($3 - 3.5$ $\mu$atm yr$^{-1}$). Although slightly smaller our results broadly agree with trend estimates based on measurements of 4.6 - 6.1 $\mu$atm yr$^{-1}$ over 2008-2015

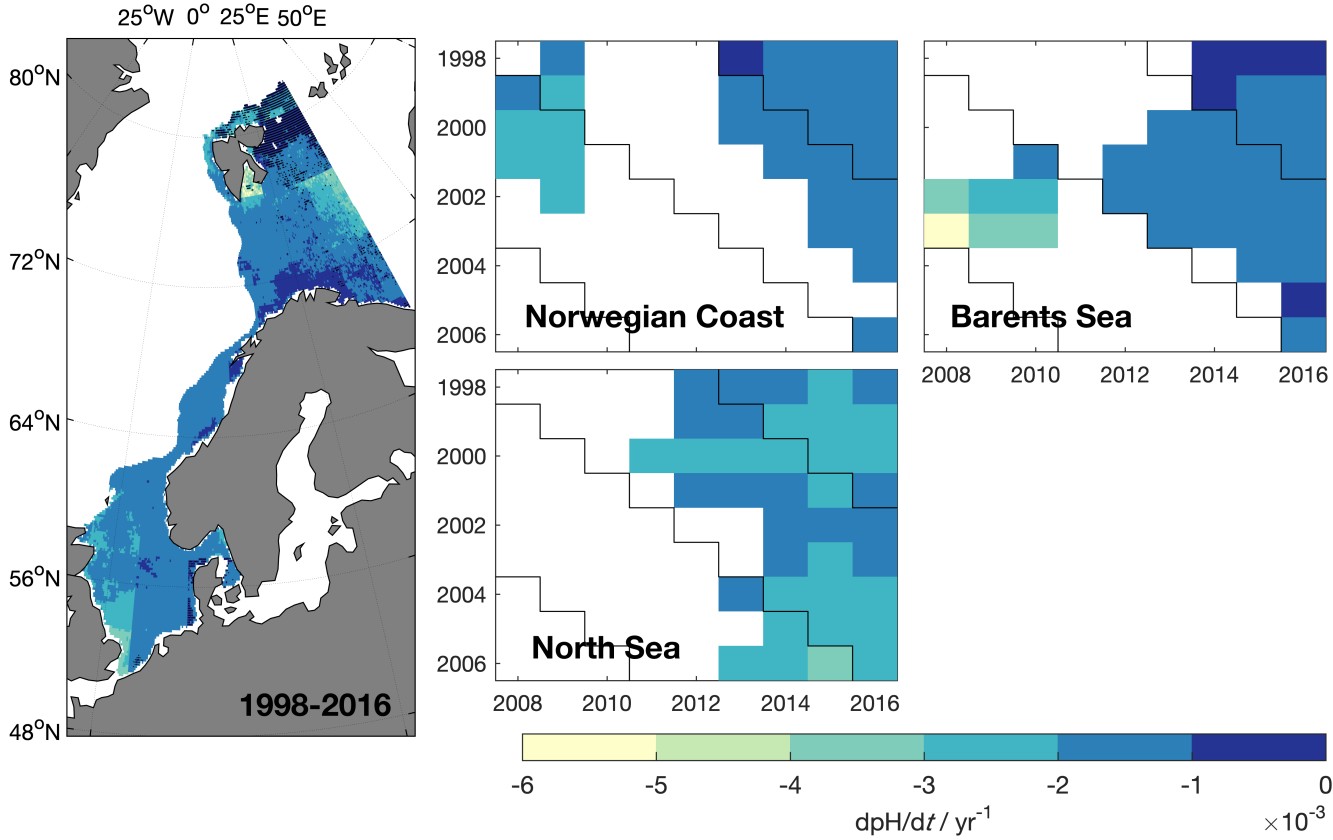

**Figure 11.** The trend in surface ocean pH estimated from deseasonalized pH. On the left hand the spatial distribution of the trend over the time period from 1998 to 2016 is shown. Grid boxes without a significant trend are denoted with a black dot. The three right hand panels show the trends in different time periods in three regions, from the various years on the y-axis to the various years on the x-axis. Non significant trends were left blank.

(Schneider and Müller, 2018). Finally, it also needs to be noted that the uncertainty of the $f$CO$_2$ maps was highest in the Baltic Sea. This makes it also more difficult, if not impossible, to properly detect these small differences in the trends.

For pH, the trend in most regions is around -0.002 yr$^{-1}$ (Figure 11). As expected, regions with the strongest trend in $f$CO$_2$ also show the highest trend in pH, such as the southern North Sea. The trend in the northern North Sea and along the Norwegian Coast is in good agreement with the pH trends found in studies focusing on the open Atlantic Ocean (-0.0022 yr$^{-1}$ (Lauvset and Gruber, 2014)) and the North Atlantic and Nordic Seas (-0.002 yr$^{-1}$ (Lauvset et al., 2015)).

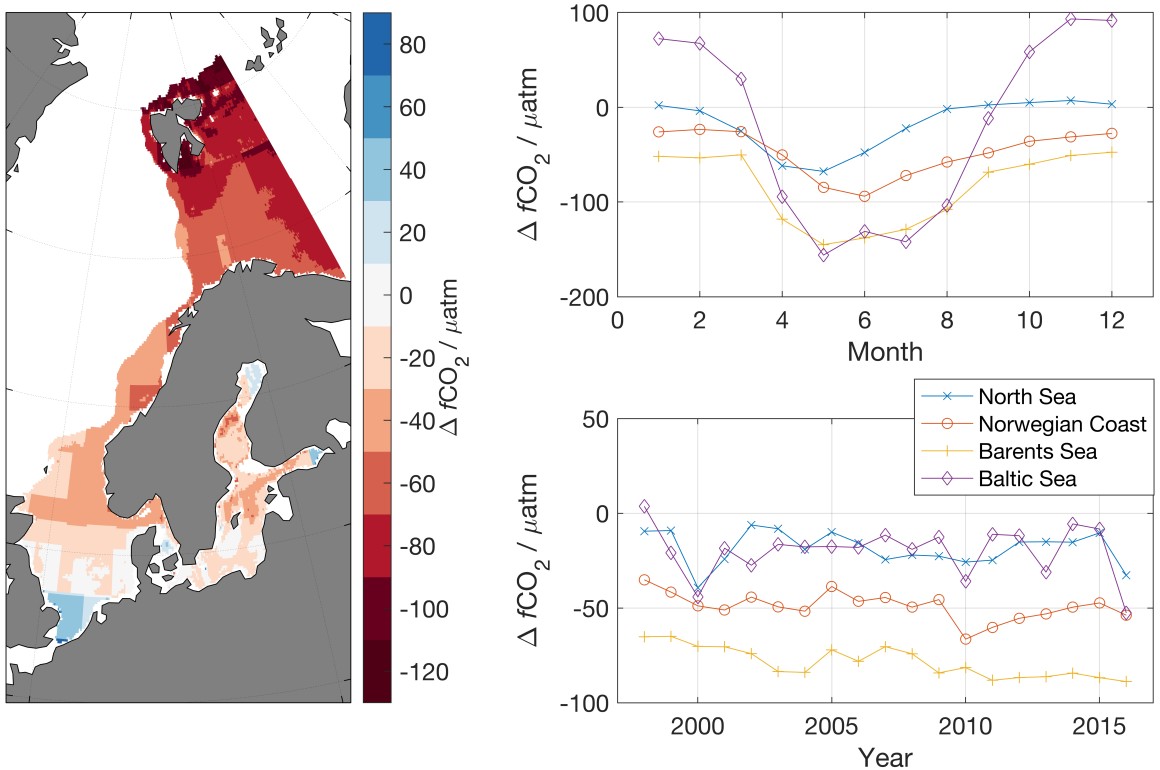

**Figure 12.** The average air-sea $CO_2$ disequilibrium over the period 1998-2016 (left hand panel, red colors indicate average undersaturation, while blue colors indicate average oversaturation). For every region average disequilibria are shown as seasonal averages (right side, upper corner) and time-series of annual disequilibria (right side, lower corner). Blue line: North Sea, red line: Norwegian coast, yellow line: Barents Sea, purple line: Baltic Sea

### 4.3 $CO_2$ disequilibrium and flux

The average air-sea $CO_2$ disequilibrium ($\Delta f CO_2 = f CO_{2,\mathrm{sea}} - f CO_{2,\mathrm{atm}}$) is shown in Figure 12. The only region showing an average supersaturation is the southern North Sea. Towards the north, the surface ocean becomes more and more undersaturated, with lowest values in the Barents Sea. The values in the Barents Sea (-60 to -80 $\mu$atm in the southern Barents Sea and less

5 than -100 $\mu$atm around Svalbard) are in agreement those estimated by Yasunaka et al. (2018). The seasonal cycle of $\Delta f CO_2$ follows a biologically driven pattern with higher values in the winter and lower values from April to August. The seasonal cycle is largest in the Baltic and smallest in the Barents Sea.

The air-sea $CO_2$ fluxes and their trends (Figure 13) show that most regions are a net and increasing sink for $CO_2$. The only net source regions are the southern North Sea and the Baltic Sea. The two different regimes in the North Sea with the southern,

10 nonstratified part being a source and the northern temporarily stratified part a sink for $CO_2$, have been described in the literature

before (Thomas et al., 2004), but the gradient between them as represented here, may be a too sharp (Section 3.1). However, there is a large interannual variability in the $f\mathrm{CO_2}$ disequilibrium (Omar et al., 2010) and studies based on different years find conflicting results regarding the direction of the flux (Kitidis et al., 2019; Schiettecatte et al., 2007; Thomas et al., 2004). This large interannual variability was also present in our maps. During some years, larger parts of the North Sea were a net source, while during other years also the southern North Sea acted as net sink (not shown).

The seasonal variations in the air-sea flux are driven by a combination of the changes in the disequilibrium, the wind strength, and the ice cover. As there is less wind during summer, when the disequilibrium is large, but a smaller disequilibrium during winter, when the wind strength is high, the seasonal variability in the flux is often less clear than that in the disequilibrium. This can be seen in the Barents Sea and Norwegian Coast. Yasunaka et al. (2018) found the seasonal and interannual variation in the Barents Sea and the Norwegian Sea mostly corresponded to the wind speed and the sea ice concentration. We see the strongest dependence on the air-sea disequilibrium, however (not shown). This indicates that the seasonal and interannual variability in our $f\mathrm{CO_2}$ maps is larger than in the maps generated by Yasunaka et al. (2018). Still, our average fluxes fit well with those reported by Yasunaka et al. (2018) of -8 to -12 mmol m$^{-2}$ d$^{-1}$ (Barents Sea) and -4 to -8 mmol m$^{-2}$ d$^{-1}$ (Norwegian Coast). In the North Sea there is almost no net flux during winter, as the surface ocean is more or less in equilibrium with the atmosphere. In the Baltic Sea, there are high fluxes into the atmosphere during winter as here a large oversaturation coincides with strong winds. This is also why the Baltic Sea is a net source region. Although Parard et al. (2017) did find slightly smaller seasonal fluxes (+15 mmol m$^{-2}$ d$^{-1}$ during winter and -8 mmol m$^{-2}$ d$^{-1}$ during summer), the annual air-sea CO$_2$ fluxes are in good agreement (0 to +4 mmol m$^{-2}$ d$^{-1}$ between 1998 and 2011).

The uncertainty in the calculated fluxes is a result of the uncertainties in the $f\mathrm{CO_2}$ observations, $\Delta f\mathrm{CO_2}$ maps, the gas exchange parameterization and the wind product. The uncertainty of the $\Delta f\mathrm{CO_2}$ is mostly driven by the uncertainty of the MLR, resulting in an error between 12 $\mu$atm and 39 $\mu$atm, according to the RMSE values of MLR1 for the different regions (Table 5). A number of studies addresses the uncertainty of gas exchange parameterizations and the wind products (Couldrey et al., 2016; Gregg et al., 2014; Ho and Wanninkhof, 2016). For this study, we apply an uncertainty of the gas transfer velocity of 20% (Wanninkhof, 2014). This will result in an uncertainty of the air-sea flux of about 2 mmol C d$^{-1}$ m$^{-2}$. It has to be kept in mind, that the absolute uncertainty in k increases with increasing wind speed, but that the uncertainty in the wind speed has largest influence in summer when also the disequilibrium is large. In contrast, the uncertainty in $\Delta f\mathrm{CO_2}$ will cause larger errors in winter, when the wind speeds are high.

There is an ongoing discussion, how and to which extent the dominant climate mode in the North Atlantic, the North Atlantic Oscillation (NAO) is driving the variability in the CO$_2$ fluxes (Salt et al., 2013; Tjiputra et al., 2012; Watson et al., 2009). Even though some features in the time series seem to coincide with very extreme states of the NAO, such as a very large disequilibrium along the Norwegian Coast in 2010, we could not find any significant correlation between the CO$_2$ fluxes and the NAO index.

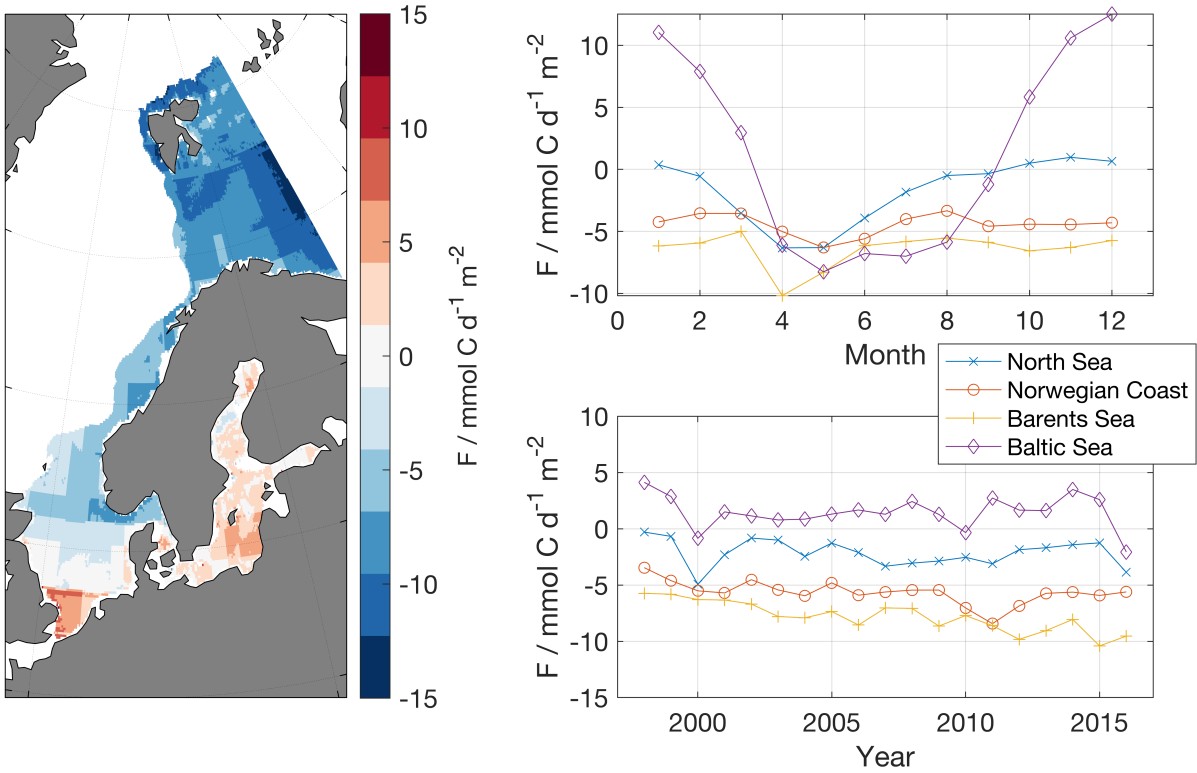

**Figure 13.** The average air-sea $CO_2$ flux over the period 1998-2016 (left hand panel, red colors indicate sink regions, while blue colors indicate source regions). For every region average fluxes are show as seasonal averages (right side, upper corner) and timeseries of annual fluxes (right side, lower corner).

## 5  Conclusions

The MLR approach presented in this work is a relatively easy and straight forward method to produce monthly $f\mathrm{CO}_2$ maps with a high spatial resolution in coastal seas, and the use of available open ocean maps improved the coastal maps significantly. The maps reproduce nicely the main spatial and temporal patterns that present in observations in the different regions for both $f\mathrm{CO}_2$ and pH. The surface seawater $f\mathrm{CO}_2$ trends were mostly lower than the atmospheric trends and also lower than the trends found in the open North Atlantic. Results show that the northern European shelf is an increasing net sink for $\mathrm{CO}_2$. Only the Baltic Sea is a net source region. This method clearly has the potential to be extended to a larger region. However, it should be handled with care in regions with only a small number of observations as the MLR can lead to unrealistic values.

Longterm observations with a high temporal resolution are extremely important for developing maps such as presented here. While a decent spatial coverage exists for the open North Atlantic, most coastal regions are still undersampled, in particular relative to their high variability in time and space. To further understand and interpret the trends in $f\mathrm{CO}_2$ and pH it is necessary

to increase our knowledge and understanding of the interaction between primary production, respiration in the water column and the sediments, mixing and gas exchange and their influence on the carbon cycle.

Besides an increased amount of in-situ observations, also improved, higher resolution driver data hold the potential to enable a better representation of spatial gradients. Including not only satellite derived chlorophyll data, but also CDOM, can also lead to an improved performance of the regressions, especially in regions with a high load of terrestrial dissolved organic carbon.

While MLR derived sea surface $f\text{CO}_2$ maps provide a coherent picture of the entire region, they have clear limitations and should be interpreted with caution in regions with few or none observations. Both, for producing high quality maps, as well for their validation a large number of observations is essential.Also, observations of a second parameter of the carbon system would be beneficial for deriving pH maps; to reduce and quantify the error introduced by estimating alkalinity from salinity. In addition to that, our work neglects the areas closest to land due to unavailability of $\text{CO}_2$ data and reanalysis products in those areas. For adding their contribution to the flux estimates, new platforms specialized on measurements directly at the land-ocean interface need to be developed.

*Data availability.* The dataset is available under: https://doi.org/10.18160/939X-PMHU.

**Appendix A: Trend in surface ocean $f\text{CO}_2$ observations**

*Competing interests.* The authors declare no competing interests.

*Acknowledgements.* First of all, we want to thank everyone involved in the collection and quality control of surface ocean $\text{CO}_2$ data. The Surface Ocean $\text{CO}_2$ Atlas (SOCAT) is an international effort, endorsed by the International Ocean Carbon Coordination Project (IOCCP), the Surface Ocean Lower Atmosphere Study (SOLAS) and the Integrated Marine Biosphere Research (IMBeR) program, to deliver a uniformly quality-controlled surface ocean $\text{CO}_2$ database. The many researchers and funding agencies responsible for the collection of data and quality control are thanked for their contributions to SOCAT. We used NCEP Reanalysis 2 data provided by the NOAA/OAR/ESRL PSD, Boulder, Colorado, USA, from their web site at https://www.esrl.noaa.gov/psd/. This study has been conducted using E.U. Copernicus Marine Service Information. This research was funded by the Research Council of Norway projects ICOS-Norway (Grant 245927) and Nansen Legacy (Grant 276730); the VERIFY project (European Union's Horizon 2020 research and innovation program grant agreement No 776810); and BONUS Integral.

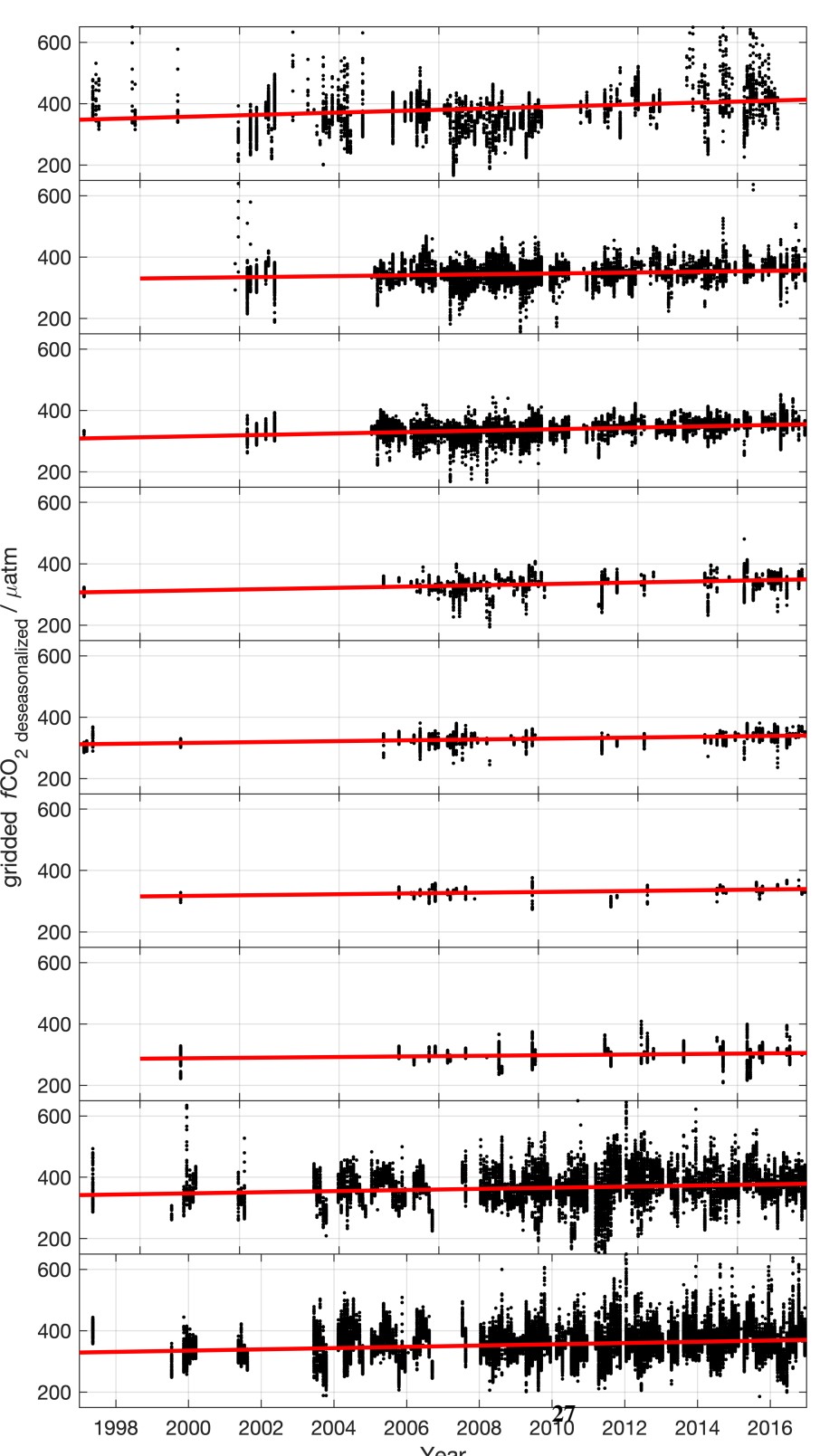

North Sea, South
51$^{\circ}$N - 54.5$^{\circ}$N
(3.2±1.3) $\mu$atm yr$^{-1}$

North Sea, Center
54.5$^{\circ}$N - 58$^{\circ}$N
(1.43±0.21) $\mu$atm yr$^{-1}$

North Sea, North
58$^{\circ}$N - 62$^{\circ}$N
(2.320±0.089) $\mu$atm yr$^{-1}$

Norwegian Coast, South
62$^{\circ}$N - 68$^{\circ}$N
(2.12±0.19) $\mu$atm yr$^{-1}$

Norwegian Coast, North
68$^{\circ}$N - 73$^{\circ}$N
(1.426±0.099) $\mu$atm yr$^{-1}$

Barents Sea, South
69$^{\circ}$N - 74$^{\circ}$N
(1.31±0.30) $\mu$atm yr$^{-1}$

Barents Sea, North
74$^{\circ}$N - 85$^{\circ}$N
(1.01±0.22) $\mu$atm yr$^{-1}$

Baltic Sea, South
54$^{\circ}$N - 56$^{\circ}$N
(2.05±0.12) $\mu$atm yr$^{-1}$

Baltic Sea, North
56$^{\circ}$N - 61$^{\circ}$N
(1.84±0.21) $\mu$atm yr$^{-1}$

• SOCAT observations
— Linear regression

gridded $fCO_2$ deseasonalized / $\mu$atm

Year

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
