# Peer review of "The northern European shelf as increasing net sink for CO2"

_Biogeosciences, 2019_

## Referee Comment (RC1) · Anonymous Referee #1 · 6 Feb 2020

MAJOR COMMENTS

The authors use a MLR approach applied to the SOCAT CO2 data-base to reconstruct a spatially and temporally resolved data-set from 1998 to 2018 in the European continental shelf. From this data-set the authors analyze the temporal trends of pCO2 during winter in different regions (North Sea, Baltic Sea, Norwegian coast & Barents seas) that are compared to the increase of atmospheric CO2.

A more detailed and in-depth analysis could be made. For instance, the authors compute the trends based on winter-only data. However, since they have a fully seasonally resolved reconstructed data-set, they could also analyze the temporal trends using summer-only data. Are the trends the same ? In addition, they could compute the trends using the full annual average, which in principle should provide the most robust

estimate of inter-annual variations since it integrates all components of seasonal variations. Are the results for the full annual average the same as the winter-only or the summer-only trends?

The other question that the authors could attempt to address is how useful is this MRL approach compared the raw SOCAT data-set to compute temporal trends. So, would the analysis of temporal trends of the raw SOCAT data give the same results as the MRL expanded data-set ? Of course this would require to aggregate the raw data into larger boxes (for instance 3 large boxes for the North Sea: southern bight of the North Sea, Central North Sea and Northern North Sea) to overcome the lower coverage of the raw SOCAT data. This question is motivated by the fact that the European Shelf is one of the areas which is most dense in CO2 data, so that you need to address the question of the usefulness of using a complex MRL approach to reconstruct and gap-fill for an original data-set that is one of the most dense for continental shelves.

Figure 9 shows that in the Southern bight of the North Sea (<53°N) there's a very strong difference between the part along the UK coast (red color = strong increase of pCO2 in time) and the part along the Dutch coast (blue color = very low increase of pCO2 in time). The two regions are clearly separated along a line that seems to correspond approximately to the 2° meridian. This line seems to also separate the Central and Northern North Sea although the differences in pCO2 trends are not as marked. But this is really strange as the spatial pCO2 distributions in the Southern Bight of the North Sea are relatively homogeneous horizontally (Thomas et al. 2004; Schiettecatte et al. 2007) so it's really odd that the temporal trends should be so different. This seems to be related to the way the MRL was implemented in the North Sea that seems to have been divided into East and West regions (along the 2° meridian) in the computation scheme (I guess). Anyway this needs to be addressed, either change the computation scheme to avoid this spatial artefact, or if this is "real" then please provide an explanation for this odd looking spatial difference.

MODERATE COMMENTS

P2 L9 : "small currents caused by the topography" does not cover the full spectrum and complexity of physical processes in continental shelves. In continental shelves there are difference buoyancy sources (thermal and haline stratification) and mixing processes (tides, upwelling, internal waves) that lead to contrasted physical settings. Please refer to classical paper by Blanton (1991).

P2 L5-14 : The introduction on the differences between coastal and open ocean waters seems to miss some important elements. CO2 patterns in costal environments are more complex that in the open ocean because overall coastal waters are more productive than open ocean, because there are several sources of nutrients such as mixing processes at continental margins (upwelling and internal wave mixing) and riverine-estuarine inputs. In addition shallow areas are vertically mixed while deeper areas are seasonally stratified. Please refer to classical paper of Wollast (1998). Overall this leads to important spatial heterogeneity and strong horizontal gradients of productivity that are reflected in equivalent gradients in surface CO2.

P2 L 15 : Please briefly explain why methods for open ocean are inadequate for coastal waters and provide references if available.

P 3 L 20 : define "winter season" in the southern north sea diatom blooms can start as early as February.

P8 L 13 Nondal et al. (2009) report a TA-salinity relation for the Northern North Atlantic Ocean that should be applicable for Norwegian coast and Barents sea but it could useful to check if it is applicable in the North Sea (e.g. Salt et al. 2013), and in particular in the Southern North Sea (Hoppema et al. 1990).

P 13 L 9-10 : Calling this comparison "validation" is a bit surprising. The authors used the SocatV5 data to generate a fCO2 data using MLR and then compare it again to the original SocatV5 data. This is not a real validation.

P14L8 you discuss data in 2017 and 2018 but at the end of the introduction (P4L3) you

say that you look at trends from 1998 to 2016.

P18L4: Paper of Sharples covers the period 1974 and 2003, so it's a stretch to assume that the trend for the 1974-2003 was continued over the period of 1998 to 2016. There are several other papers that have addressed recent changes of phytoplankton phenology in the North Sea.

P20L6: "The lower trend stems most likely from an earlier onset of spring bloom" The authors have the data to test this, since they have reconstructed a temporally resolved data-set. If the onset of the bloom is earlier in the year, then so should the peak of the bloom. The seasonal $CO_2$ minimum is a good proxy for peak spring phytoplankton, so the authors can check if this has changed in time and occurred earlier in the year.

P20L24: "The sea-air $CO_2$ fluxes (Figure 12) show that most regions are a net and increasing sink for $CO_2$. The only source net regions are the southern North Sea and the Baltic Sea. The two different regimes in the North Sea with the southern, nonstratified part being a source and the northern temporarily stratified part a sink for $CO_2$, are well described in the literature (Thomas et al., 2004)." Thomas et al. (2004) only sampled the North Sea during 4 cruises, and their "spring" cruise was in mid-May, when the spring phytoplankton in the Southern Bight of the North Sea is over. So Thomas et al. (2004) missed the peak of the spring bloom (and minimum of $CO_2$) that occurs in April, as clearly shown by the work of Schiettecatte et al. (2007) and Omar et al. (2010). This is why Thomas et al. (2004) reported the Southern Bight of the North Sea as a source of $CO_2$ to the atmosphere, since their data-set does not represent the period of strong $CO_2$ under-saturation during spring. The better seasonally resolved data-set of Schiettecatte et al. (2007) shows in fact that the Southern Bight of the North Sea is a small sink of atmospheric $CO_2$, although admittedly lower than the Northern North Sea.

MINOR COMMENTS

The text contains several typos and inadequate terminology.

P 2 L 5 : terms like coastal seas, coastal seas or continental shelves would be more adequate than "coasts"

P8 L 12 : "calculating ocean acidification" is an awkward expression. You calculated pH from which you compute a trend. This trend is not necessarily negative (acidification). In some coastal areas an increase of pH has been reported, in other areas there is no trend (Duarte et al. 2013).

P 8 L16: "river moths" => river mouths

P19L4: "eutrification" => eutrophication

Legend of Figure 4. Is incorrect. The figures show deltafCO2 not fCO2

P17L8 : "to validate this to validate this"

P19L5 : Can you provide a reference showing the effect of eutrophication on CO2 ?

REFERENCES

Blanton JO (1991)Circulation processes along oceanic margins in relation to material fluxes. In Ocean Margin Processes in Global Change, Mantoura RFC, Martin J-M, Wollast R (eds) Wiley

Duarte et al. (2013) Is ocean acidification an open-ocean syndrome? Understanding anthropogenic impacts on seawater pH. Estuaries and Coasts 36(2), DOI: 10.1007/s12237-013-9594-3

Hoppema J.M.J. (1990) The distribution and seasonal variation of alkalinity in the Southern Bight of the North Sea and in the Western Wadden Sea, Netherlands Journal of Sea Research, 26, 11-23 Omar et al. (2010) Spatiotemporal variations of fCO2 in the North Sea, Ocean Sci., 6, 77-89.

Salt, L. A et al. (2013), Variability of North Sea pH and CO2 in response to North Atlantic Oscillation forcing, J. Geophys. Res. Biogeosci., 118,

doi:10.1002/2013JG002306.

Schiettecatte et al. (2007) High temporal coverage of carbon dioxide measurements in the Southern Bight of the North Sea, Marine Chemistry, 106, 161–173

Thomas, H., Bozec, Y., Elkalay, K., de Baar, H.J.W., 2004. Enhanced open ocean storage of CO2 from shelf sea pumping. Science 304, 1005–1008

Wollast, R. (1998) The global coastal ocean, in The Sea, edited by K. H. Brink and A. R. Robinson, pp. 213– 252, John Wiley, New York.

---

## Referee Comment (RC2) · Anonymous Referee #2 · 18 Mar 2020

This is an interesting manuscript that tackles an important problem: maps interpolating sparse observations of surface ocean pCO$_2$ (and related variables like pH) perform well in the open ocean but generally do not accurately reproduce the conditions seen in more complex shelf sea environments like the northwest European continental shelf, the focus of this study. The authors apply a long-established technique (MLR) but with the innovative step of using low-resolution open-ocean pCO$_2$ maps as one of the predictors. They tested two different open-ocean pCO$_2$ maps and also developed a 'traditional' MLR based only on other in situ variables. One of the open-ocean maps, which did project pCO$_2$ values across the shelf seas, performed slightly better than the traditional MLR but the other, which did not, performed better or worse depending on the metric considered, although the authors state it was better. The former

open-ocean-map-based MLR was therefore used to derive most of the results. The discussion is mostly a description of the trends in surface ocean $pCO_2$, air-sea $CO_2$ fluxes and pH in the relevant shelf seas.

There are a few issues I think the authors should consider revising before publication:

One of the strongest reasons to use an MLR instead of a neural network approach is the relative ease with which the predictive model can be shared and used by other researchers. Please would the authors therefore provide the actual fitted coefficients to their equation 1.

The word 'coasts' is used throughout to describe the study area but it is not clear how this is defined. For me 'coast' would refer to the very near coastal zone (e.g. intertidal areas) as opposed to 'shelf sea' which would go out to a depth contour of e.g. 200 m. The results do not also extend all the way to the coast, as can be seen from the white gaps between land and ocean on Figures 4, 5, and 9–12 and noted in the penultimate sentence of the Conclusions. Please explicitly define, and consider revising, the terminology used.

Is it valid to predict all the way up into the northern Baltic Sea given that there appears to be only one month of data there (Figure 2)?

The previous study results given in Table 1 for the North Sea show a range of different values (specifically, Thomas et al. (2007) vs Salt et al. (2013)) and also covering different time periods, with Salt et al. finding a different rate of change from 2001-2005 compared with 2005-2008. Salt et al. implicate the NAO as a key driver of this short-term variability, but this study does not mention the NAO explicitly. Do these new results provide any evidence for the NAO influencing air-sea $CO_2$ exchange here? On the other hand, Figure 9, upper left grid box panel for the North Sea, indicates that no significant trend can be found in the North Sea for these short periods reported by previous studies. Implicitly, this figure is therefore saying that the different trends reported in previous studies are in fact not significant. Is that a point the authors intend

to make? Either way it feels like there is some interesting discussion missing here.

p19, line 1 states the western North Sea did not show a significant trend, but this area does not have black dots in Figs 9 and 10. Are trends significant here or not? Also, this paragraph as a whole does not effectively justify or explain its opening sentence.

Please provide details of all CO2SYS options selected (e.g. borate:chlorinity). Consider using the newer CO2SYS v2 from Orr et al. (2018) and including error propagation from the equilibrium constants in your calculations?

Finally, a few minor points to consider:

It is noted several times that and old version of SOCAT (v5) was used for the fitting before the explanation on p8 that the reason for this was so that the newer version could be used to independently test the fits. It would be helpful to mention this the first time SOCAT is discussed. Why do the different panels in Figure 3 (in particular the second panel) show different subsets of SOCAT data points?

Figure 4: colour bar should be labelled $fCO_2$, not $\Delta fCO_2$.

Figures 5, 9, etc.: maps contain a lot of straight lines and right angles, usually indicates boundaries between regions with different predictive equations but they don't entirely match with the regions shown in Figure 1, what is the cause?

Figure 9: what is the difference between a cross and a circle?

The colour scale on Figure 11 feels counterintuitive, as usually $CO_2$ source areas are shown in red and sinks in blue.

p9 line 2: missing citation.

p10 line 3: MLR, not MLD.

In units for rates please explicitly clarify whether d means decade or day.

There are a few issues with the English language throughout so this aspect should also

be carefully checked through.

I support the comments and suggestions made by the other reviewer.

**0.1 References**

Orr, J. C., Epitalon, J.-M., Dickson, A. G. and Gattuso, J.-P.: Routine uncertainty propagation for the marine carbon dioxide system, Mar. Chem., 207, 84–107, doi:10.1016/j.marchem.2018.10.006, 2018.

Salt, L. A., Thomas, H., Prowe, A. E. F., Borges, A. V., Bozec, Y. and de Baar, H. J. W.: Variability of North Sea pH and $CO_2$ in response to North Atlantic Oscillation forcing, J. Geophys. Res. Biogeosci., 118(4), 2013JG002306, doi:10.1002/2013JG002306, 2013.

Thomas, H., Prowe, A. E. F., Heuven, S. van, Bozec, Y., Baar, H. J. W. de, Schiettecatte, L.-S., Suykens, K., Koné, M., Borges, A. V., Lima, I. D. and Doney, S. C.: Rapid decline of the $CO_2$ buffering capacity in the North Sea and implications for the North Atlantic Ocean, Global Biogeochem. Cy., 21(4), doi:10.1029/2006GB002825, 2007.

---

## Author Comment (AC1) · 20 May 2020

Dear Reviewer 2,

We want to thank reviewer 2 for their constructive comments and questions which we carefully addressed. Please see our detailed responses below (reviewer comment in italics, author response, change in manuscript in gray)

Kind regards, Meike Becker

Please also note the supplement to this comment:
https://www.biogeosciences-discuss.net/bg-2019-480/bg-2019-480-AC1-supplement.pdf

[Figure]

[Figure]

**Supplement:**

*(reviewer comment,* author response, change in manuscript)

*The authors use a MLR approach applied to the SOCAT CO2 data-base to reconstruct a spatially and temporally resolved data-set from 1998 to 2018 in the European continental shelf. From this data-set the authors analyze the temporal trends of pCO2 during winter in different regions (North Sea, Baltic Sea, Norwegian coast & Barents seas) that are compared to the increase of atmospheric CO2.*
*A more detailed and in-depth analysis could be made.*

*For instance, the authors compute the trends based on winter-only data. However, since they have a fully seasonally resolved reconstructed data-set, they could also analyze the temporal trends using summer-only data. Are the trends the same ?*
*In addition, they could compute the trends using the full annual average, which in principle should provide the most robust estimate of inter-annual variations since it integrates all components of seasonal variations. Are the results for the full annual average the same as the winter-only or the summer-only trends?*

We do not use winter-only data for estimating the trends. The trends shown in this manuscript are computed over the entire year. We did compute also winter only and summer only trends for comparison to the other studies. Trends in summer were generally less significant than the all-year or winter-only trends. We agree with Referee#1 that the contribution of the different seasons to the overall trend is an interesting feature that can be investigated further. We therefore added a paragraph addressing the trend in pCO2 for every month.

The hypothesis, that an earlier or more intense bloom onset is responsible for the relatively low trends in the North Sea is supported by looking at the contributions of the different months to the overall trend. Figure 10 show the trend for each month in the four different regions.

[Figure]

*Figure 1 The trend in surface ocean fCO$_2$ estimated resolved per month (1998 to 2016).*

*The other question that the authors could attempt to address is how useful is this MRL approach compared the raw SOCAT data-set to compute temporal trends. So, would the analysis of temporal trends of the raw SOCAT data give the same results as the MRL expanded data-set ? Of course this would require to aggregate the raw data into larger boxes (for instance 3 large boxes for the North Sea:*

*southern bight of the North Sea, Central North Sea and Northern North Sea) to overcome the lower coverage of the raw SOCAT data. This question is motivated by the fact that the European Shelf is one of the areas which is most dense in CO2 data, so that you need to address the question of the usefulness of using a complex MRL approach to reconstruct and gap-fill for an original data-set that is one of the most dense for continental shelves.*

Reviewer 1 is right, when they state, that the European shelfis one of the coastal regions in the world with highest density in CO2 data, especially when looking at the northern North Sea and parts of the Baltic Sea. However, that is not similarly true for all European shelf regions. In the North Sea it would definitely be worthwhile to perform a proper data-based, high resolution trend analysis for the entire basin and then comparing the results to ours. We think, that our manuscript here is not the right place to do so. For the northern part of the North Sea, there is a recent study by (Omar et al., 2019) focusing on winter trends. They find the same trends as we: no significant trend east of about 5E and a trend close to the atmospheric trend west of 5E. We followed also the suggestion of RW1 and performed a quick-and-dirty trend analysis for 9 large boxes (based on deseasonalized gridded SOCAT data from SOCAT v5). The results of this analysis support the results from our maps. We added a table with the SOCAT-based trends to the fCO2 trend section of manuscript and a picture showing the regression analysis to the supplement.

Principally, we do think, that is there is a large value in developing gap filling methods also in regions with a high data density. The major application of gap filled pCO2 products lays not the estimation of trends in pCO2, but in air-sea CO2 fluxes and estimating the ocean carbon sink. For this pCO2 data covering all months, years and regions is crucial.

The northern European shelf is a region with a high data density. In order to validate the general patterns of $f$CO$_2$ trends we estimated the $f$CO$_2$ trends also from the SOCAT v5 observations, that was used to produce the MLR (Table 6). We gridded and deseasonalized the SOCAT v5 data and divided the entire region into 9 subregions. A figure showing the fits and the data coverage can be found in Appendix A.

These directly observation based trends show similar general patterns as those based on our maps (Figure 8, 1998-2016): (1) largest trends in the southern North Sea, (2) decreasing towards the North with trends around the atmospheric trend in the northern North Sea and trends around 1 µatm yr$_{-1}$ in the Barents Sea, (3) close to atmospheric trends in the Baltic Sea.

| Region | Latitude / °N | Trend / $\mu$atm yr$^{-1}$ |
|---|---|---|
| North Sea, South | 51 - 54.5 | $3.2 \pm 1.3$ |
| North Sea, Center | 54.5 - 58 | $1.43 \pm 0.21$ |
| North Sea, North | 58 - 62 | $2.320 \pm 0.089$ |
| Norwegian Coast, South | 62 - 68 | $2.12 \pm 0.19$ |
| Norwegian Coast, North | 68 - 73 | $1.426 \pm 0.099$ |
| Barents Sea, South | 69 - 74 | $1.31 \pm 0.30$ |
| Barents Sea, North | 74 - 85 | $1.01 \pm 0.22$ |
| Baltic Sea, South | 54 - 56 | $2.05 \pm 0.12$ |
| Baltic Sea, North | 56 - 61 | $1.84 \pm 0.21$ |

*Table 1 fCO$_2$ trend calculated from gridded, deseasonalized SOCAT v5 observations.*

[Figure]

*Figure 2: Trend in surface ocean fCO₂ in deseasonalized, gridded observation data (SOCAT v5).*

*Figure 9 shows that in the Southern bight of the North Sea (<53_N) there's a very strong difference between the part along the UK coast (red color = strong increase of pCO2 in time) and the part along the Dutch coast (blue color = very low increase of pCO2 in time). The two regions are clearly separated along a line that seems to correspond approximately to the 2_ meridian. This line seems to also separate the Central and Northern North Sea although the differences in pCO2 trends are not as marked. But this is really strange as the spatial pCO2 distributions in the Southern Bight of the North Sea are relatively homogeneous horizontally (Thomas et al. 2004; Schiettecatte et al. 2007) so it's really odd that the temporal trends should be so different. This seems to be related to the way the MRL was implemented in the North Sea that seems to have been divided into East and West regions (along the 2_ meridian) in the computation scheme (I guess). Anyway this needs to be addressed, either change the computation scheme to avoid this spatial artefact, or if this is "real" then please provide an explanation for this odd looking spatial difference.*

These lines are a remnant of the open ocean pCO2 maps, which were used as a driver in the MLR (in this case Rödenbeck, 4x5˚ resolution).

> As most the driver data has a smaller resolution than the final maps (see Table3) the grid of the driver data is still visible in the final maps. This is specifically the case for the used open ocean $p$CO$_2$ maps. Residuals of the original open ocean Rödenbeck map (resolution 5 x 4) are clearly visible in the MLR 1 maps as well as the trends and fluxes calculated from these.

MODERATE COMMENTS

*P2 L9 : "small currents caused by the topography" does not cover the full spectrum and complexity of physical processes in continental shelves. In continental shelves there are difference buoyancy sources (thermal and haline stratification) and mixing processes (tides, upwelling, internal waves) that lead to contrasted physical settings. Please refer to classical paper by Blanton (1991).*

We agree that in coastal regions more diverse physical processes involved. The processes we named was meant as examples. However, we changed the sentence and added a reference to Blanton (1991).:

> Small scale circulation patterns governed by topographic features, thermal and haline stratification, or mixing though tidal cycles, upwelling or internal waves result in a need for more complex maps with a higher resolution (Bricheno et al., 2014; Lima et al., 2012; Blanton, 1991)

*P2 L5-14 : The introduction on the differences between coastal and open ocean waters seems to miss some important elements. CO2 patterns in costal environments are more complex that in the open ocean because overall coastal waters are more productive than open ocean, because there are several sources of nutrients such as mixing processes at continental margins (upwelling and internal wave mixing) and riverine estuarine inputs. In addition shallow areas are vertically mixed while deeper areas are seasonally stratified. Please refer to classical paper of Wollast (1998). Overall this leads to important spatial heterogeneity and strong horizontal gradients of productivity that are reflected in equivalent gradients in surface CO2.*

We agree completely with RW1 that the description of differences between open ocean and coastal regions was lacking some of the biogeochemical characteristics. We changed the paragraph accordingly:

> Generally, coastal regions show a larger productivity than open ocean regions due to different additional sources of nutrients (e.g. mixing at continental margins, river runoff). While deeper regions are seasonally stratified, shallow regions are vertically mixed allowing for exchange between the benthic and pelagic parts of the ecosystem (Griffiths et al.,2017, Wollast,1998). Together with strong gradients of productivity this leads to spatial and temporal heterogeneity in surface $CO_2$ content.

*P2 L 15 : Please briefly explain why methods for open ocean are inadequate for coastal waters and provide references if available.*

We do not state that the methods are not suitable for coastal oceans. The work of (Laruelle et al., 2017) for example is based on the SOM-FFN method of (Landschützer et al., 2017). Our point is that the currently existing open ocean maps have a too scarce resolution and therefore cannot be used in coastal regions (This is stated in the text). Recently, many reanalysis products that are used as driver variables became available in a higher resolution. In addition to that, the computers get stronger and stronger. This now enables the production of maps with a higher resolution. (At least in regions with sufficient pCO2 observations)

*P 3 L 20 : define "winter season" in the southern north sea diatom blooms can start as early as February.*

We added the information, which months were used in the respective literature studies to Table 1.
We discuss at different places throughout the manuscript that the variability of spring bloom start, especially in coastal regions, is one of the major limitations of using winter-only trend estimates in coastal regions. In the new version of the manuscript there will be a new paragraph in the discussion section focusing the influence of the season on the trend estimate.

*P8 L 13 Nondal et al. (2009) report a TA-salinity relation for the Northern North Atlantic Ocean that should be applicable for Norwegian coast and Barents sea but it could useful to check if it is applicable in the North Sea (e.g. Salt et al. 2013), and in particular in the Southern North Sea (Hoppema et al. 1990).*

We agree with RW1 that using the (Nondal et al., 2009) equation results in larger uncertainties in the North Sea, especially the southern North Sea than at the Norwegian Coast or in the Barents Sea. Using Nondal et al equation in the North Sea will most likely result in underestimating the alkalinity at low salinities. In the North Sea the low salinity water usually come either from the Baltic Sea of riverine input. The regressions shown in (Salt et al., 2013) and (Omar et al., 2019) show a larger intercept than the equation we used. Both regressions are focusing on the Skagerrak region and the Northern North Sea  The work of (Hoppema, 1990) in the Southern North Sea and the Wadden Sea shows in general very high alkalinity vs salinity ratios compared to the other two studies.
Besides the Skagerrak, regions with low salinity can be found close to shore and at the big river mouths. When looking on the GLODAP dataset alkalinity and salinity in these regions is not correlated at all. Therefore, the pH in these regions should be handled with care. However, for the majority of the North Sea where the salinity is varying between 34 and 35, the equation of Nondal et al describes the salinity-alkalinity correlation better than the equations of Salt et al and Omar et al, which are based on the Baltic Sea inflow. We agree, that using different equations for the different regions could be a good way to improve pH maps based on fCO2 maps in the future.

*P 13 L 9-10 : Calling this comparison "validation" is a bit surprising. The authors used the SocatV5 data to generate a fCO2 data using MLR and then compare it again to the original SocatV5 data. This is not a real validation.*

We do compare our maps against independent data. We predict our maps for the years 2017 and 2018 and compare these against SOCAT data from these years (SOCATv2019). However, as this obviously is not stated clearly enough, we changed the introduction of this section.

> The prediction of the maps into the years 2017 and 2018 will be compared with data from the newest SOCAT release (SOCATv2019) to have a comparison with an independent dataset.

We added also a sentence to the data handling section:

A newer version of the SOCAT database (SOCATv2019) was used for validating the maps against independent data.

We acknowledge that the naming of the section 'Validation' caused confusion and changed its name to 'Performance'.

*P14L8 you discuss data in 2017 and 2018 but at the end of the introduction (P4L3) you say that you look at trends from 1998 to 2016.*

We use the data from 2017 and 2018 for validating our maps. The trends are calculated only until 2016. We did not calculate trends for the latter years, as these have a higher uncertainty due data from these years not being included in the fits. For extending the fits to 2018, we recommend obtaining a new fit equation that includes data from 2017 and 2018. The general goal of these maps is an annual release of the maps based on the latest SOCAT version.

*P18L4: Paper of Sharples covers the period 1974 and 2003, so it's a stretch to assume that the trend for the 1974-2003 was continued over the period of 1998 to 2016. There are several other papers that have addressed recent changes of phytoplankton phenology in the North Sea.*

We added (Desmit et al., 2020) as a reference. The paragraph was changed to:_

The bloom timing and onset in the North Sea after the 1990s has been shown to be mainly triggered by the spring-neap tidal cycle and the air temperature (Sharples et al., 2006). The bloom timing and onset was found to be significantly earlier in the 2010s compared to the previous decades (Desmit et al., 2019).

*P20L6: "The lower trend stems most likely from an earlier onset of spring bloom" The authors have the data to test this, since they have reconstructed a temporally resolved data-set. If the onset of the bloom is earlier in the year, then so should the peak of the bloom. The seasonal CO2 minimum is a good proxy for peak spring phytoplankton, so the authors can check if this has changed in time and occurred earlier in the year.*

We tested this and found the low trends to come mainly from spring (see Figure above). When plotting the seasonal cycles for pCO2 in the early part of the time series in comparison to the later part of the time series, there is a shift to an earlier decrease in pCO2 during spring visible. We think that this is a very interesting topic and it certainly holds the potential for further, more detailed investigation. However, we also think that this will go beyond the aim of this manuscript.

*P20L24: "The sea-air CO2 fluxes (Figure 12) show that most regions are a net and increasing sink for CO2. The only source net regions are the southern North Sea and the Baltic Sea. The two different regimes in the North Sea with the southern, nonstratified part being a source and the northern temporarily stratified part a sink for CO2, are well described in the literature (Thomas et al., 2004)." Thomas et al. (2004) only sampled the North Sea during 4 cruises, and their "spring" cruise was in mid-May, when the spring phytoplankton in the Southern Bight of the North Sea is over. So Thomas et al. (2004) missed the peak of the spring bloom (and minimum of CO2) that occurs in April, as clearly shown by the work of Schiettecatte et al. (2007) and Omar et al. (2010). This is why Thomas et al. (2004) reported the Southern Bight of the North Sea as a source of CO2 to the atmosphere, since their data-set does not represent the period of strong CO2 under-saturation during spring. The better seasonally resolved data-set of Schiettecatte et al. (2007) shows in fact that the Southern Bight of the North Sea is a small sink of atmospheric CO2, although admittedly lower than the Northern North Sea.*

We do not agree that different timing is the reason why (Schiettecatte et al., 2007) reports the southern North Sea as a sink for CO2, while (Thomas et al., 2004) find it being a source. All spring cruises in (Schiettecatte et al., 2007) were very late in the month. 11BE20040329 and 11BA20040524 are in the SOCAT database. As the paper states that there was a time difference of 28 days between the March,

April and May cruises, respectively, we can assume that the April cruise also took place during the last days of April. That means that the May cruise in (Thomas et al., 2004) (64PE2002506) started only a week after Schiettecatte et al.'s April cruise. (Thomas et al., 2004) might have missed the minimum, but we doubt that this effect is large enough. We think the difference in the various flux estimates is largely driven by interannual variability. As you can see from the data presented in (Omar et al., 2010) bloom timing and intensity can vary rapidly from year to year. Another large factor in comparing these flux estimates is the used wind velocities. Both studies (Schiettecatte et al., 2007 and Thomas et al., 2004) use wind velocity during the time of the cruise. This means the in Schiettecatte et al., (2007) wind data from a few days in the end of the month is used for reporting a monthly flux.
We extended the discussion about this point:

> The sea-air $CO_2$ fluxes show that most regions are a net and increasing sink for $CO_2$. The only source net regions are the southern North Sea and the Baltic Sea. The two different regimes in the North Sea with the southern, nonstratified part being a source and the northern temporarily stratified part a sink for $CO_2$, have been described in the literature before (Thomas et al., 2004). However, there is a large interannual variability in the f $CO_2$ disequilibrium (Omar et al., 2010). This is reflected in the fact that studies based on different years find conflicting results regarding the direction of the flux (Schiettecatte et al, 2007, Thomas et al., 2004). This large interannual variability can also be found in our maps. During some years larger parts of the North Sea were a net source, while during other years also the southern North Sea acted as net sink.

MINOR COMMENTS
*The text contains several typos and inadequate terminology.*

We carefully read through the text again and corrected it.

*P 2 L 5 : terms like coastal seas, coastal seas or continental shelves would be more adequate than "coasts"*

We went through the text and changed the general term coasts to coastal seas or continental shelves

*P8 L 12 : "calculating ocean acidification" is an awkward expression. You calculated pH from which you compute a trend. This trend is not necessarily negative (acidification). In some coastal areas an increase of pH has been reported, in other areas there is no trend (Duarte et al. 2013).*

We changed 'calculating ocean acidification' to 'calculating pH'.

*P 8 L16: "river moths" => river mouths*

changed

*P19L4: "eutrification" => eutrophication*

changed

*Legend of Figure 4. Is incorrect. The figures show deltafCO2 not fCO2*

changed

*P17L8 : "to validate this to validate this"*

corrected

*P19L5 : Can you provide a reference showing the effect of eutrophication on CO2 ?*

Added references here

**References:**

*Desmit, X., Nohe, A., Borges, A.V., Prins, T., Cauwer, K.D., Lagring, R., Zande, D.V. der, Sabbe, K., 2020. Changes in chlorophyll concentration and phenology in the North Sea in relation to de-eutrophication and sea surface warming. Limnology and Oceanography 65, 828–847. https://doi.org/10.1002/lno.11351*

*Hoppema, J.M.J., 1990. The distribution and seasonal variation of alkalinity in the Southern Bight of the North Sea and in the Western Wadden Sea. Netherlands Journal of Sea Research 26, 11–23. https://doi.org/10.1016/0077-7579(90)90053-J*

*Landschützer, P., Gruber, N., Bakker, D.C.E., 2017. An updated observation-based global monthly gridded sea surface pCO2 and air-sea CO2 flux product from 1982 through 2015 and its monthly climatology (NCEI Accession 0160558). Version 2.2.*

*Laruelle, G.G., Landschützer, P., Gruber, N., Tison, J.-L., Delille, B., Regnier, P., 2017. Global high-resolution monthly $pCO_2$ climatology for the coastal ocean derived from neural network interpolation. Biogeosciences 14, 4545–4561. https://doi.org/10.5194/bg-14-4545-2017*

*Nondal, G., Bellerby, R.G.J., Oldenc, A., Johannessena, T., Olafssond, J., 2009. Optimal evaluation of the surface ocean CO2 system in the northern North Atlantic using data from voluntary observing ships. Limnol. Oceanogr. 7, 109–118.*

*Omar, A.M., Olsen, A., Johannessen, T., Hoppema, M., Thomas, H., Borges, A.V., 2010. Spatiotemporal variations of f CO2 in the North Sea. Ocean Sci. 13.*

*Omar, A.M., Thomas, H., Olsen, A., Becker, M., Skjelvan, I., Reverdin, G., 2019. Trends of Ocean Acidification and pCO2 in the Northern North Sea, 2003–2015. Journal of Geophysical Research: Biogeosciences 124, 3088–3103. https://doi.org/10.1029/2018JG004992*

*Salt, L.A., Thomas, H., Prowe, A.E.F., Borges, A.V., Bozec, Y., Baar, H.J.W. de, 2013. Variability of North Sea pH and CO2 in response to North Atlantic Oscillation forcing. Journal of Geophysical Research: Biogeosciences 118, 1584–1592. https://doi.org/10.1002/2013JG002306*

*Schiettecatte, L.-S., Thomas, H., Bozec, Y., Borges, A.V., 2007. High temporal coverage of carbon dioxide measurements in the Southern Bight of the North Sea. Marine Chemistry, Special issue: Dedicated to the memory of Professor Roland Wollast 106, 161–173. https://doi.org/10.1016/j.marchem.2007.01.001*

*Thomas, H., Bozec, Y., Elkalay, K., Baar, H.J.W. de, 2004. Enhanced Open Ocean Storage of CO2 from Shelf Sea Pumping. Science 304, 1005–1008. https://doi.org/10.1126/science.1095491*

---

## Author Comment (AC2) · 20 May 2020

Dear Reviewer 2,

We want to thank reviewer 2 for their constructive comments and questions which we carefully addressed. Please see our detailed responses below (reviewer comment in italics, author response, change in the manuscript in gray).

Kind regards, Meike Becker

Please also note the supplement to this comment:
https://www.biogeosciences-discuss.net/bg-2019-480/bg-2019-480-AC2-supplement.pdf

[Figure]

**Supplement:**

*(reviewer comment,* author response, change in manuscript)

*This is an interesting manuscript that tackles an important problem: maps interpolating sparse observations of surface ocean pCO2 (and related variables like pH) perform well in the open ocean but generally do not accurately reproduce the conditions seen in more complex shelf sea environments like the northwest European continental shelf, the focus of this study. The authors apply a long-established technique (MLR) but with the innovative step of using low-resolution open-ocean pCO2 maps as one of the predictors. They tested two different open-ocean pCO2 maps and also developed a 'traditional' MLR based only on other in situ variables. One of the open-ocean maps, which did project pCO2 values across the shelf seas, performed slightly better than the traditional MLR but the other, which did not, performed better or worse depending on the metric considered, although the authors state it was better. The former open-ocean-map-based MLR was therefore used to derive most of the results. The discussion is mostly a description of the trends in surface ocean pCO2, air-sea CO2 fluxes and pH in the relevant shelf seas.*

*There are a few issues I think the authors should consider revising before publication: One of the strongest reasons to use an MLR instead of a neural network approach is the relative ease with which the predictive model can be shared and used by other researchers. Please would the authors therefore provide the actual fitted coefficients to their equation 1.*

We recommend strongly to develop a specific fit for any new application. The fit coefficients are expected to change with using different products of driving data. Additionally, the driving data products are updated and improved regularly. As these annual updates often also involve changes in the historic data a new fit should be done for any new combination of driver data.

*The word 'coasts' is used throughout to describe the study area but it is not clear how this is defined. For me 'coast' would refer to the very near coastal zone (e.g. intertidal areas) as opposed to 'shelf sea' which would go out to a depth contour of e.g. 200 m. The results do not also extend all the way to the coast, as can be seen from the white gaps between land and ocean on Figures 4, 5, and 9–12 and noted in the penultimate sentence of the Conclusions. Please explicitly define, and consider revising, the terminology used.*

We changed the use of the very general term coasts to coastal seas or continental shelves. The definition for coastal seas as used in this work can be found under Methods: Study area. A limiting factor for the extension of the maps to land is the availability of driver data. Intertidal areas for example are not represented in the driver data. We also added a sentence in the section 'Methods: Study area' to clarify what we mean when we use the term coastal seas in this manuscript.

Please note, that this study concentrates on the continental shelf area. the near coastal zones (e.g. intertidal zones) are not included due to the limited availability of driver data in these regions.

*Is it valid to predict all the way up into the northern Baltic Sea given that there appears to be only one month of data there (Figure 2)?*

This is a good question. One could easily argue to remove this part. We decided to include it to give a flux estimate for the entire Baltic Sea. Within ICOS, there was a new underway pCO2 system installed on a commercial vessel sailing through the Gulf of Bothnia in 2019. We therefore expect a much better data coverage in the region from 2019 on. It will be very interesting to compare the maps we show here with an updated version that include a full annual cycle in the Gulf of Bothnia.

*The previous study results given in Table 1 for the North Sea show a range of different values (specifically, Thomas et al. (2007) vs Salt et al. (2013)) and also covering different time periods, with Salt et al. finding a different rate of change from 2001- 2005 compared with 2005-2008. Salt et al. implicate the NAO as a key driver of this short-term variability, but this study does not mention the NAO explicitly. Do these new results provide any evidence for the NAO influencing air-sea CO2 exchange*

*here? On the other hand, Figure 9, upper left grid box panel for the North Sea, indicates that no significant trend can be found in the North Sea for these short periods reported by previous studies. Implicitly, this figure is therefore saying that the different trends reported in previous studies are in fact not significant. Is that a point the authors intend to make? Either way it feels like there is some interesting discussion missing here.*

We think that in depth testing of underlying drivers, such as NAO, is exceeding the aim of this manuscript. Here, we primarily want to present the maps. That being said, we did of course have a look at potential driving factors, but we did not find evidence for the NAO to be a key driver in any of the regions. When looking into detail there are a few features that seem to be related (such as for example the large disequilibrium in the Norwegian Coast region in 2010, a year with a very negative NAO index).

*p19, line 1 states the western North Sea did not show a significant trend, but this area does not have black dots in Figs 9 and 10. Are trends significant here or not? Also, this paragraph as a whole does not effectively justify or explain its opening sentence.*

We changed this sentence to:

> The observation that large subareas (the Baltic Sea, along the shore of the western North Sea) did not show a significant trend can be explained by the fact, that coastal sea systems, especially enclosed areas as the Baltic Sea, experience a high anthropogenic pressure.

*Please provide details of all CO2SYS options selected (e.g. borate:chlorinity). Consider using the newer CO2SYS v2 from Orr et al. (2018) and including error propagation from the equilibrium constants in your calculations?*

Added the information about the boron-salinity ratio. We are working on including the error propagation into our scripts and this will be included in a future, updated release of the maps.

Finally, a few minor points to consider:
*It is noted several times that and old version of SOCAT (v5) was used for the fitting before the explanation on p8 that the reason for this was so that the newer version could be used to independently test the fits. It would be helpful to mention this the first time SOCAT is discussed.*

The possibility to compare was one point and the other was the time that past when preparing and analyzing the maps. We added the following sentence to the 'data handling' section:

> A newer version of the SOCAT database (SOCATv2019) was used for validating the maps against independent data.

*Why do the different panels in Figure 3 (in particular the second panel) show different subsets of SOCAT data points?*

changed

*Figure 4: colour bar should be labelled fCO2, not _fCO2.*

changed

*Figures 5, 9, etc.: maps contain a lot of straight lines and right angles, usually indicates boundaries between regions with different predictive equations but they don't entirely match with the regions shown in Figure 1, what is the cause?*

These lines are an artifact stemming from the open ocean pCO2 maps that were used as a driver. You can see here the remains of the 4x5 ˚ grid of the original Rödenbeck et al product.

*Figure 9: what is the difference between a cross and a circle?*

Significant increase/decrease of temperature with time. This is described both in the figure itself, as in the figure description.

*The colour scale on Figure 11 feels counterintuitive, as usually $CO_2$ source areas are shown in red and sinks in blue.*

Changed the color code in Figure 11 and 12

*p9 line 2: missing citation.*

Added reference

*p10 line 3: MLR, not MLD.*

changed

*In units for rates please explicitly clarify whether d means decade or day.*

The unit of all rates shown in this manuscript is per year. It is the unit of the fluxes is per day. We do not see the need for clarification here.

*There are a few issues with the English language throughout so this aspect should also be carefully checked through.*
*I support the comments and suggestions made by the other reviewer.*

---

## Author Response (AR2)

Reviewer comments in gray italics, my answer in black and the changes in the text in "…" and bold

**Answer to referee 1**

*The authors have partly addressed my previous comments. However, two important points were eluded.*

*First, the authors acknowledge that there is an **artifact in the interpolation scheme** that led to strange CO2 patterns in the North Sea, and an artificial difference between the west and the east side of the North Sea in particular in the SBNS that has been shown in the past as relatively homogeneous. They acknowledge that "These lines are a remnant of the open ocean pCO2 maps, which were used as a driver in the MLR (in this case Rödenbeck, 4x5˚ resolution)." However, they did little to try to correct this by adjusting/modifying the interpolation scheme. I do not see the added value of interpolating data and using fancy MRL approaches in an area where the data coverage is already very dense, to provide in the end clearly biased maps that the authors did not bother to try to correct.*

We regret that we have missed to address all the previously raised concerns by the referee. It was certainly not our intention to elude the important points raised. While we do understand and agree (hence our acknowledgement in the text) with the referee's argument that the coarse resolution nature of the open ocean driver data (namely the Rödenbeck 4x5 degree pCO2 field) and the resulting "patchy" pCO2 reconstruction appears too heterogenous in contrast to other studies. The referee therefore suggests correcting our approach in order to improve performance, however there are some noteworthy complications that prevent us from doing so.

Firstly, the construction of the input data for our MLR is out of our hand. While we would benefit from a finer resolved Mixed Layer interpolation scheme by Rödenbeck et al, (and equally finer resolved physical proxies such as temperature and salinity, etc) this is not feasible considering the built-up of the method. That said, Rödenbeck et al have now created a finer resolved version of their scheme (with 2.5˚ x 2˚ resolution), however, even with this finer resolved version we still see this spatial gradient. Other products with higher resolution that extend to the coast and may serve as an alternative for the future are currently in review (see also Laruelle et al and the product of Landschützer et al 2020 and discussion below), but none of these alternative products offers a resolution attempted in our study. Our primary intention is to make use of the coarse resolution existing pCO2 estimates to provide novel and fine-resolved coastal estimates, but not to improve the existing estimates. That said, we believe this still is valuable information.

Secondly, we believe it behooves us well to highlight shortcomings of our approach, even when they are outside our ability to change. That said, we agree that this deserves a more detailed discussion than previously provided.

Thirdly, we agree with the argument by the referee that direct measurements do not show these gradients as strongly and are therefore more reliable. Nevertheless, for many applications, such as model evaluation or the investigation of regional trends a high-resolution gap-filled pCO2 product is required, desired or even inevitable. In addition, the amount of

available data is not such that they can be mapped with confidence every single year. We have first-hand experience, and gaps due to instrumental failure and funding issues, do occur. Here we offer a first, though not bias-free, estimate that aims to be applied to all coastal regions of the western Nordic Seas, discussing its shortcomings and offer ways forward to improve it in the future. One way forward would be to improve the resolution of the open ocean pCO2 product. A second possible way forward would be to apply different drivers, as we potentially do not need pCO2 as a driver for data rich regions. Illustrating and discussing ways forward is certainly something we have missed in our previous manuscript and our first response to the referee's concerns, however, working out the technical aspects is, as we still believe, beyond the scope of this study. In this study we focus on the best and most robust scheme for all regions combined.

Therefore, considering the point raised by the referee and our answers above, we have extended the discussion and added the following:

In the Results (p 10, l 11 – p 12, l 6):

**"We notice that the gradients that exist between the grid cells in the Rödenbeck map, are still visible in our maps in some regions, for example the sharp gradient in the southern North Sea in February, or the east-west and north-south gradients in the entire North Sea in August. Such gradients are also evident in directly mapped pCO$_2$ data (Kitidis et al. 2009), however, here they are more strongly meridional and latitudinal in their extent. As such, while these gradients do reflect actual features of the pCO$_2$ distribution in the North Sea, their specific shape here, are also a consequence of the influence of the Rödenbeck maps on our estimates; from the use of these maps as a driver in the MLR and their importance in improving the statistical performance vs the MLR that did not use these values as a driver (MLR 1 vs MLR 3, Table 5). Also, they do reflect the uncertainty of - and our level of confidence in - the estimated pCO$_2$ values; being approximately similar to or slightly larger than the RMSE of MLR 1 (Table 5). Any smoothing would be completely artificial, and, while being more visually pleasing, would not better reflect the truth in any meaningfully quantifiable extent. We have therefore chosen to leave them untouched. These gradients are therefore also visible in subsequent pH and trend maps."**

In the Discussion (p 14, l 9 – l 7):

**"One clear drawback of the here presented MLR 1 is the clearly visible grid-pattern of the open ocean pCO$_2$ product that was used as input data with its grid size of 5° x 4°. This artifact implies sharper gradients in fCO$_2$ than can be found in observations. There are two ways how one could get rid of this artifact in a future release. A finer resolution of the used open ocean maps will lead to a better representation of the actual gradients in our mapped product. Rödenbeck et al. just released a newer, finer resolution of their open ocean product that we intend to use in a future version of this data product. Additionally, running the MLR without an open ocean pCO$_2$ product can provide a coastal pCO$_2$ product without this artifact (given that all other driving parameters, such as temperature or mixed layer depth, also are available in the required resolution). While in principle it is preferential to have coastal maps that are independent of the open ocean**

**products, MLR 3, which is running without open ocean pCO$_2$ as driver, did not reach the same accuracy as MLR 1. New and better input fields or a different regression method could help improving the independent coastal maps in the future."**

*Second, the authors acknowledge that the pCO2 shows a shift in the spring bloom timing. This is a really interesting result that would strongly contribute to the future impact of this work. So I do not understand that the authors did not include this in the paper, I do not see how this could be "outside the scope of this work" as replied.*

Again, it was certainly not our intention to elude the important points raised and agree that this observation should be included and explored to a larger extent in the discussion. We do believe, however, it is "out of scope" for us to quantify the reason why this shift has happened, as this would entail detailed examination of the atmospheric, oceanic and ecosystem conditions that can bring about such a change. We also note, that the monthly resolution of our maps somewhat restricts abilities to detect changes in the timing of the onset of the spring bloom, as such changes may be a matter of days to weeks. That being said we do see a significant shift the bloom timing in the western North Sea between the first and the second half of our time series. We added a panel showing the average pCO2 seasonalities in the northwestern North Sea from 1998 to 2007 and 2007 to 2016. We also added extended discussion and added an additional panel to Figure 10.

(p 20, l 18 – p 21, l 6)

**"Figure 10a shows the annual trends in fCO$_2$ in each month in the four regions considered. Particularly in the North Sea and Baltic, very low fCO$_2$ trends are observed in February – May, suggesting that changing timing of the spring bloom might be important here. Investigating the seasonal fCO$_2$ in more detail (Figure 10b), revealed an earlier and deeper fCO$_2$ drawdown in the second decade of our analysis (2007-2016) than in the first (1998-2007) in the northeastern North Sea (58 – 60˚N, 3 -8˚E). This strongly suggest that an earlier and stronger spring bloom is lowering the annual pCO$_2$ growth rates in this region, which is among the ones with the smallest fCO$_2$ trends (X $\mu$atm yr$^{-1}$, Fig. 9). In the other regions, no such changes could be established with confidence. Future investigations should aim at generating fCO$_2$ maps with higher temporal resolution, as changes in the timing of the spring bloom might be a matter of days or weeks, which would not be fully resolved by the monthly maps presented here."**

[Figure]

**Figure 10: (a)The trend in surface ocean fCO$_2$ estimated resolved per month (1998 to 2016). (b) The average seasonality in fCO$_2$ for the periods 1998-2007 (green) and 2007-2016 (purple) in the northeastern North Sea (58 – 60˚N, 3 -8˚E), normalized to December. The standard deviation for each month is shown as shaded area.**

*Can the authors add to the discussion how their approach compares to the recent work of Landschützer et al. (2020) that seem to provide a consistent and uniform interpolation scheme for the open and coastal oceans.*

At the time of initial submission, the study by Landschützer et al 2020 was not yet submitted and the submission of the 1$^{st}$ revision the study only existed as a pre-print (i.e. has not undergone peer review in its online form). To follow rigorous scientific standards, we try and avoid discussing grey literature. Fortunately, the study by Landschützer et al 2020 has now been accepted for publication. We understand the resemblance between these studies and the resulting need for clarification. There are several major differences between the study of Landschützer et al 2020 and this work:

Firstly, Landschützer et al 2020 do not provide a new estimate in our chosen study domain, but combines the open ocean estimate by Landschützer et al 2016 with the coastal ocean estimate by Laruelle et al 2017. Therefore, most regions that are both covered in our study and in the Landschützer et al 2020 estimate actually stem from Laruelle et al 2017, which we do discuss in our manuscript.

Secondly, Landschützer et al 2020 combine their estimates to provide a 0.25x0.25 degree climatology covering the global ocean. Here, we, on the one hand, provide a higher resolution local estimate, which on the other hand focuses on longer term signals rather than seasonal variations.

To acknowledge the existence of this climatology, and its potential to further improve our local high-resolution approach, we added to the text:

(p 2, l 22 – l 24)

> **"A global climatology covering both open ocean and coastal regions was recently**

**performed by combining this product with the open ocean product of Landschützer et al (2016) (Landschützer et al.,2020).”**

*Refs*

*Landschützer, P., Laruelle, G. G., Roobaert, A., and Regnier, P.: A uniform pCO2 climatology combining open and coastal oceans, Earth Syst. Sci. Data Discuss., https://doi.org/10.5194/essd-2020-90, in review, 2020.*

Reviewer comments in gray italics, my answer in black and the changes in the text in "…" and bold

**Answer to referee 2**

*Many of the issues raised in the first review have been well addressed but the authors have argued that they do not wish to address others.*

*The unwillingness to publish the actual MLR equation used remains a concern. The given reason is "We recommend strongly to develop a specific fit for any new application." What if the new application is to directly compare the result here with another MLR - to compare the directions and relative importance of the variables driving the fit - either developed for the same region with newer data or for a different region? Not showing the different terms also hinders assessment of whether the fit makes sense, whether terms with well-known relationships with pCO2 (e.g. temperature) are driving it in the expected direction, whether the relationships are meaningful and linked to underlying processes (and therefore probably more reliable for predicting beyond the input dataset) or simply spurious correlations for the specific training data used. Finally, keeping something so critical to the study as a secret "black box" seems to contradict the open access principles of this journal.*

We regret that we have missed to address all the important points by raised the referee during the first revision. It was certainly not our intention to hide the MLR coefficients, but our first intuition was, that the equations are strongly optimized for the chosen study domain, hence we wanted to avoid the impression we have created a generalized MLR model (that may be applied to other regions as well). In spite of our initial intention we may indeed have missed the opportunity to be transparent and instead presented a black box. That being said, we think that the use of the MLR coefficients for interpretation is limited because (a) many drivers (e.g. chlorophyll) serve as proxies for higher order processes, (b) many drivers have manifold and sometimes opposite influence on the carbonate system (e.g. temperature effects both the gas solubility as well as the Schmidt number but can act as a proxy for upwelled cold deep water) which makes a direct interpretation of the coefficients challenging. However, we as mentioned above, we certainly don't want to hide the equations, hence we now added the equations of the MLR to the supplementary.

We added the following to the manuscript:

(p 9, l 19 – l 20)

**«The coefficients for MLR 1, MLR 2 and MLR 3 are provided in the supplement. «**

*The comparison between previous results in Table 1 vs the significance plot in Figure 9 has not been addressed. The authors may of course decide that discussing the actual drivers (e.g. NAO) of interannual variability is beyond the scope of this study. But even ignoring*

*the drivers, the point remains that the new results here show that trends determined over the short periods that were the basis of previous studies are not significant. As things stand, a new result that implicitly casts doubt on earlier work is being presented here without comment.*

We are sorry that we did not address the concern of incompletely discussing our results in a satisfactory way. In the following we want to give a detailed answer to both discussion points raised by the referee.

Regarding the actual drivers of the shown interannual variability, we do think that this topic easily can get very extensive and is better addressed in a separate manuscript, particularly in light of multiple climate modes dominating on various timescales (see e.g. Landschützer et al 2019, GRL, "Detecting regional modes of variability in observation-based surface ocean $pCO_2$) and potential teleconnections (see, e.g. Steinman, B. A., M. E. Mann, and S. K. Miller (2015), Atlantic and Pacific multidecadal oscillations and Northern Hemisphere temperatures, Science, 347, 988, doi:10.1126/science.1257856). That being said, we did test if the NAO, i.e. the climate mode previously identified as the dominant mode of variability in these regions, can explain at least some of the interannual variability that we observe. As we did not find any significant correlation over the entire study area, we decided not to concentrate further on this. However, the referee is correct in pointing out that also not finding a significant correlation is worth mentioning.

We completely agree with the referee that the non-significant $pCO_2$ trends for time ranges of less than 10 years is an important finding and needs to be stated. We deeply regret that we missed addressing this in the original manuscript.

To amend these to shortcomings we added the following two paragraphs to the manuscript:

(p 24, l 23 – l 27)

**«There is an ongoing discussion, how and to which extend the dominant climate mode in the North Atlantic, the North Atlantic Oscillation (NAO) is driving the variability in the $CO_2$ fluxes (Tjiputra et al. 2011, Salt et al, 2013, Watson et al, 2009). Even though some features in the time series seem coincident with very extreme states of the NAO, such as a very large disequilibrium along the Norwegian Coast in 2010, we could not find any significant correlation between the $CO_2$ fluxes and the NAO index.»**

(p 18, l 26 – l 30)

**«Generally, only few regressions over time ranges of less than 10 years turned out to be significant. This is an important finding when comparing the trends determined from our maps with the trends reported in literature, of which many were covering periods shorter than 10 years (Table 1). In order to compare the general patterns of $fCO_2$ determined from our maps with those directly determined from observations over a similar time range, we estimated the $fCO_2$ trends also from the SOCAT v5 observations that were used to produce the MLR (Table 6).»**

*The issue regarding the **northern Baltic Sea** containing only a single month of data was acknowledged in the author responses but it would be better to discuss this briefly in the manuscript itself. Looking back to the map on Figure 9 it's interesting to see that this region does not have black dots to indicate non-significant trends. The method therefore claims to identify a significant long-term trend for a region containing only one month from one year of training data. This makes me wonder whether significance is being assessed in a suitable way.*

We regret that we gave the referee the impression of not handling the uncertainties in an acceptable way. Figure 9 only shows the significance of the trend regression. This is independent of how much data there is to produce the MLR in the first place. We do state clearly that especially the data in the northern Baltic Sea have to be handled with care. That being said, we do understand that not marking these as questionable in Figure 9 may lead the reader to trust the trends in the Bay of Bothnia. We therefore removed this region from Figure 9.

(p 18, l 10 – l 12)

[revised manuscript text omitted]

---

## Author Response (AR3)

Reviewer comments in gray italics, my answer in black and the changes in the text in "…" and bold

Answer to referee 1:

*"The authors state that "Our primary intention is to make use of the coarse resolution existing pCO2 estimates to provide novel and fine-resolved coastal estimates". However, the use of the Rödenbeck open-ocean data in the interpolation leads to incorrect patterns in particular to an artificial west-east gradient in the North Sea. While I understand that the authors cannot resolve the problem related to the resolution of the Rödenbeck data, I do not understand how this can generate the west-east gradient in the North Sea, since the North Sea is only open to the North Atlantic in the far North, and the west-east gradient in the North Sea is observed all along the eastern coast of the UK. Does this mean that the data in the North Sea are somehow interpolated with the North Atlantic data assuming the UK is transparent? In conclusion, I fail to see to usefulness of "fine-resolved coastal estimates" that are simply wrong (bluntly artificial patterns in the North Sea) and do not compare satisfactorily with the original data."*

We see in this comment two major points, that we would like to address separately: the question about gradients in the North Sea on one hand and a technical question about how the interpolation scheme used by the Rödenbeck open ocean product effects our coastal maps on the other hand.

First of all, we want to mention that we never claimed our maps to perfectly reflect all variability. Our aim is to provide a new state-of-the-art estimate for $p\mathrm{CO}_2$ and $\mathrm{CO}_2$ flux variability over the entire North Sea. For the North Sea, we report an uncertainty of 26 µatm for $f\mathrm{CO}_2$ and 0.5µatm/yr for the trend in $f\mathrm{CO}_2$. These uncertainties are important to keep in mind when discussing possible gradients in the pictures we show in this manuscript as they are in the same order as one step in the colorbar.

Secondly, we disagree with the claim of the referee that this west-east gradient in the North Sea is entirely artificial. E.g. in the southern North Sea, studies revealed that there are different seasonal cycles of $p\mathrm{CO}_2$ reported in the literature (Voynova et al., 2018), where they found a larger amplitude in the eastern part of the North Sea than in the western part. This will lead to a longitudinal gradient at least during some parts of the year. In the northern North Sea, Omar et al. (2019) found a slightly different seasonality in $p\mathrm{CO}_2$ as well as a larger trend in $p\mathrm{CO}_2$ west of 5˚E (2.39±0.58 µatm/yr) than east of 5˚E (1.2±1.5 µatm/yr).

We think that the technical aspect of the referee's comment is very valid. Here, we first want to clarify that the position of the 5x4 grid boxes in the open ocean product is not a problem, as the grid box border passes through the 2.5˚W line, which crosses the UK almost completely on. The second aspect here is the interpolation scheme that is used to produce the Rödenbeck et al open ocean product. This indeed ignores land masses as if they are not existing. While this certainly is not perfect, we do not think that this is a big problem at this place. Firstly, the British Isles are still relatively small compared to the correlation length. Thus, the distance going around the isles presumably is not that much longer than going across the isles. Therefore, one would expect correlations between west coast and east coast, of course smaller than for a direct ocean connection. Secondly, this is a region with a lot of observations available, which reduces the effect of the correlations on the final product. Thirdly, running the MLR and fitting the Rödenbeck et al. $p\mathrm{CO}_2$, together with other driver data, to $f\mathrm{CO}_2$ observations in the North Sea will adjust for possible Atlantic influences at least partly. We believe that this is actually one

of the reasons, why we reach a much better performance with our MLR compared to the original Rödenbeck et al product.

p14, ll. 17-21

[revised manuscript text omitted]

---

## Author Response (AR4)

Dear Prof Middelburg,

We are very pleased to hear that our manuscript was accepted for publication.
We included all four corrections you suggested into the manuscript:

Page 2, l. 28 northern North Sea
Page 2, l. 35: replace on the other hand with however (because there is no on the one hand
Page 14, l. 20: European shelf
Page 26, third line from above: amount of in situ observations

With best regards,
Meike Becker

[revised manuscript text omitted]